# Estrogen receptor alpha in the brain mediates tamoxifen-induced changes in physiology in mice

Zhi Zhang[1,2], Jae Whan Park[1,2], In Sook Ahn[1], Graciel Diamante[1], Nilla Sivakumar[1,2], Douglas Arneson[1], Xia Yang[1], J Edward van Veen[1,2]*, Stephanie M Correa[1,2]*

[1]Department of Integrative Biology and Physiology, University of California Los Angeles, Los Angeles, United States; [2]Laboratory of Neuroendocrinology of the Brain Research Institute, University of California Los Angeles, Los Angeles, United States

**Abstract** Adjuvant tamoxifen therapy improves survival in breast cancer patients. Unfortunately, long-term treatment comes with side effects that impact health and quality of life, including hot flashes, changes in bone density, and fatigue. Partly due to a lack of proven animal models, the tissues and cells that mediate these negative side effects are unclear. Here, we show that mice undergoing tamoxifen treatment experience changes in temperature, bone, and movement. Single-cell RNA sequencing reveals that tamoxifen treatment induces widespread gene expression changes in the hypothalamus and preoptic area (hypothalamus-POA). These expression changes are dependent on estrogen receptor alpha (ERα), as conditional knockout of ERα in the hypothalamus-POA ablates or reverses tamoxifen-induced gene expression. Accordingly, ERα-deficient mice do not exhibit tamoxifen-induced changes in temperature, bone, or movement. These findings provide mechanistic insight into the effects of tamoxifen on the hypothalamus-POA and indicate that ERα mediates several physiological effects of tamoxifen treatment in mice.

*For correspondence:
vanveen@ucla.edu (JEV);
stephaniecorrea@ucla.edu (SMC)

**Competing interests:** The authors declare that no competing interests exist.

## Introduction

Tamoxifen is a selective estrogen receptor modulator that has been used for effective treatment of hormone responsive breast cancers for more than 40 years (*Jordan, 2003*). As an adjuvant, tamoxifen therapy can decrease the incidence of breast cancer recurrence by up to 40% (*Davies et al., 2011*). This exceptionally effective therapy remains standard of care for people with hormone-responsive cancers, and reduction of recurrence persists for at least 10 years of continuous tamoxifen treatment (*Davies et al., 2013*; *Chlebowski et al., 2014*; *Gierach et al., 2017*). In contrast to these benefits, tamoxifen therapy has been associated with a variety of negative side effects, including increased risk for hot flashes (*Love et al., 1991*; *Howell et al., 2005*; *Francis et al., 2015*), endometrial cancer and venous thromboembolic events (*Fisher et al., 1998*; *Cuzick et al., 2007*), bone loss (*Powles et al., 1996*), and fatigue (*Haghighat et al., 2003*). These responses markedly impact quality of life. Accordingly, ~25% of eligible patients fail to start or complete this life-saving therapy due to side effects and safety concerns (*Friese et al., 2013*; *Berkowitz et al., 2021*). The tissues and cells that mediate these negative side effects remain unclear. Unraveling the cells and mechanisms that mediate the positive effects of tamoxifen from those that mediate the negative side effects is necessary for understanding the multifaceted effects of tamoxifen therapy on physiology. Ultimately, this knowledge could lead to the design of new or adjuvant therapies that circumvent the side effects, improve patient quality of life, and perhaps enhance survival via increased patient compliance.

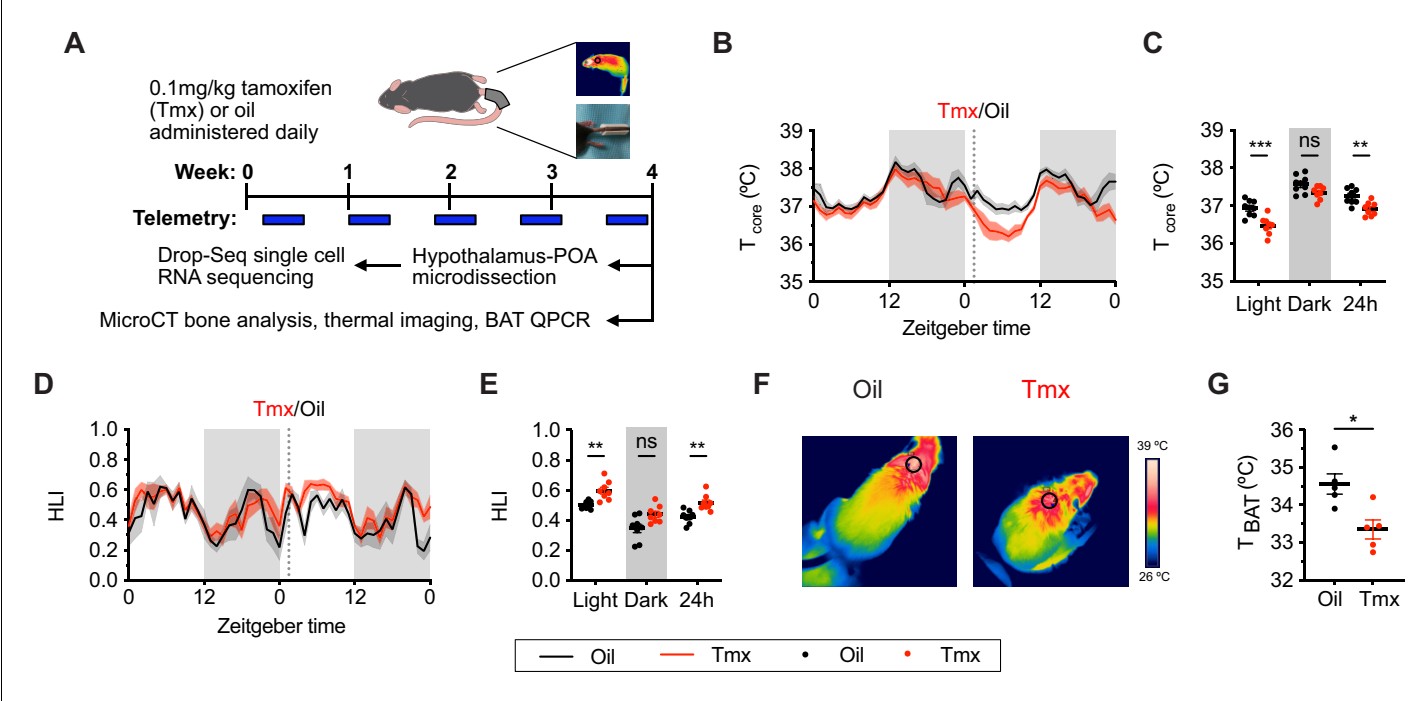

**Figure 1.** Tamoxifen treatment decreases core body temperature and increases heat dissipation. (A) Strategy to measure the physiological and gene expression effects of long-term tamoxifen (Tmx) treatment. Mice were treated by daily subcutaneous injection of 0.1 mg/kg tamoxifen or corn oil for 4 weeks. Core and tail temperature were measured every 5 min using telemetry probes. Snapshot of thermographic images for brown adipose tissue (BAT) temperature was obtained in last week of treatment. At the conclusion of 4-week treatment, experimental mice were sacrificed for bone analysis or single-cell RNA sequencing. (B) Hourly average of core body temperature over 3 days before and 3 weeks after injections (dotted line). (C) Average core body temperature from mice shown in panel (B) highlighting per animal averages in light (7:00 to 19:00), dark (19:00 to 7:00), and total 24 hr periods (n = 8, treatment: $F_{1,14}=11.92$, p=0.0039). (D) Heat loss index (HLI) calculated from continuous measurements of core and tail temperature. (E) Average HLI from mice shown in panel (D) (n = 7, treatment: $F_{1,14}=15.52$, p=0.0015). (F) Thermographic images of interscapular skin above BAT depots in mice injected with either oil or tamoxifen. (G) Quantification of temperature of skin above interscapular BAT depots (n = 5, $t_8 = 3.255$, p=0.0116). In (B and D), line shading width represents standard error of the mean (SEM). ns, not significant, *, p<0.05; **, p<0.01; ***, p<0.001 for Sidak's multiple comparison tests (C and E) following a significant effect of treatment in a two-way repeated measures ANOVA or two-tailed t-tests (G).

The online version of this article includes the following source data and figure supplement(s) for figure 1:

**Source data 1.** Source data for *Figure 1*, panel b.
**Source data 2.** Source data for *Figure 1*, panel c.
**Source data 3.** Source data for *Figure 1*, panel d.
**Source data 4.** Source data for *Figure 1*, panel e.
**Source data 5.** Source data for *Figure 1*, panel g.
**Figure supplement 1.** Tamoxifen increases temperature in tail and suppresses thermogenic gene expression in brown adipose tissue (BAT).
**Figure supplement 1—source data 1.** Source data for *Figure 1—figure supplement 1*, panel a.
**Figure supplement 1—source data 2.** Source data for *Figure 1—figure supplement 1*, panel b.
**Figure supplement 1—source data 3.** Source data for *Figure 1—figure supplement 1*, panel c.

Within the brain, the hypothalamus and preoptic area (hypothalamus-POA) is highly enriched for estrogen receptor expression and represents an excellent anatomical candidate for mediating many of the side effects of tamoxifen therapy in humans. Estrogen receptor alpha (ERα) signaling regulates body temperature (*Bowe et al., 2006*; *Musatov et al., 2007*; *Mittelman-Smith et al., 2012*; *Martínez de Morentin et al., 2014*), physical activity (*Musatov et al., 2007*; *Correa et al., 2015*; *van Veen et al., 2020*), and bone density (*Farman et al., 2016*; *Zhang et al., 2016*; *Herber et al., 2019*) through distinct neuronal populations. Indeed, the hypothalamus-POA is a demonstrated target of tamoxifen, leading to changes in food intake and body weight (*Wade and Heller, 1993*; *López et al., 2006*; *Lampert et al., 2013*) and changes in the hypothalamic-pituitary-ovarian (*Wilson et al., 2003*; *Aquino et al., 2016*) and hypothalamic-pituitary-adrenal (*Wilson et al., 2003*) axes. Tamoxifen has also been shown to affect gene expression in the hypothalamus; its

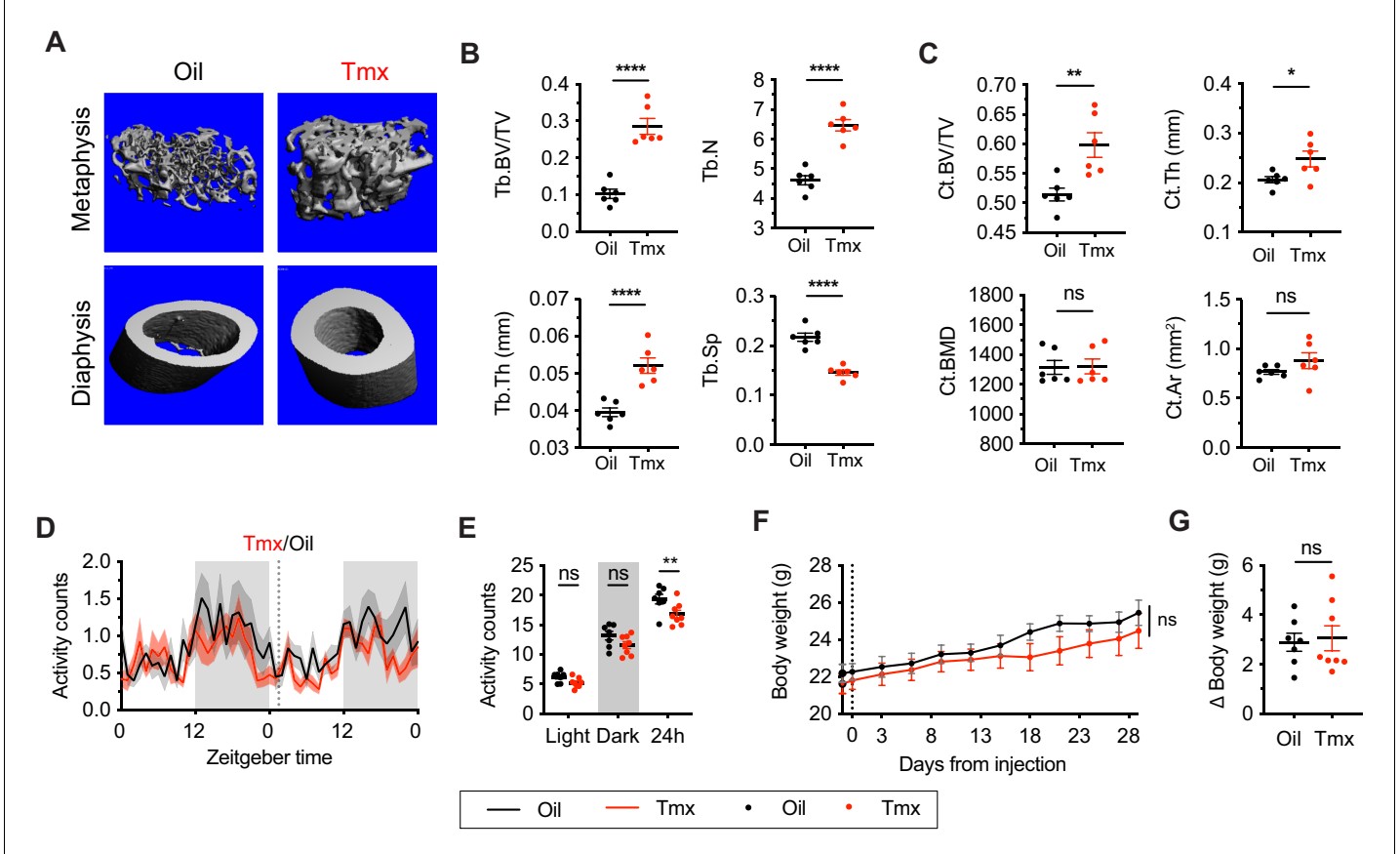

**Figure 2.** Tamoxifen treatment increases bone density and decreases movement. (**A**) Representative micro-computed tomography (microCT) images showing bone density of the distal metaphysis and midshaft diaphysis of femurs from tamoxifen or oil treated female mice. Tamoxifen or vehicle was injected subcutaneously at 0.1 mg/kg for 28 days. (**B**) Trabecular bone volume fraction (BV/TV, $t_{10} = 7.322$, p<0.0001), trabecular numbers (Tb.N, $t_{10} = 7.539$, p<0.0001), trabecular thickness (Tb.Th, $t_{10} = 5.303$, p=0.0003), and trabecular separation (Tb.Sp, $t_{10} = 7.331$, p<0.0001) in distal metaphysis of femurs (n = 6). (**C**) Bone volume fraction (Ct.BV/TV, $t_{10} = 3.557$, p=0.0052), cortical thickness (Ct.Th, $t_{10} = 2.451$, p=0.0342), bone mineral density (Ct. BMD, $t_{10} = 0.0926$, p=0.928), and cortical area (Ct.Ar, $t_{10} = 1.368$, p=0.2013) in diaphysis of femurs (n = 6). (**D**) Hourly average of movement over 3 days before and 3 weeks after tamoxifen or oil injections (dotted line) measured every 5 min using an intraperitoneal telemetry probe. Line shading width represents standard error of the mean (SEM). (**E**) Average total movement from mice shown in panel (D) (n = 8, treatment: $F_{1,14}=6.182$, p=0.0262). (**F**) Change of body weight over 28 days during tamoxifen or oil injection (n = 7–8, mixed-effects model, treatment: p=0.5468). Error bars represent SEM. (**G**) Total change in body weight over the course of the 28-day experiment (n = 8, $t_{13} = 0.2572$, p=0.8010). ns, non-significant; *, p<0.05; **, p<0.01; ***, p<0.001; ****, p<0.0001 for Sidak's multiple comparison tests (**E**) following a significant effect of treatment in a two-way repeated measures ANOVA or two-tailed t-tests (**B, C, and G**).

The online version of this article includes the following source data for figure 2:

**Source data 1.** Source data for *Figure 2*, panel b.
**Source data 2.** Source data for *Figure 2*, panel c.
**Source data 3.** Source data for *Figure 2*, panel d.
**Source data 4.** Source data for *Figure 2*, panel e.
**Source data 5.** Source data for *Figure 2*, panel f.
**Source data 6.** Source data for *Figure 2*, panel g.

administration blocks the estrogen dependent induction of the progesterone receptor (*Pgr*) in the ventromedial hypothalamus (VMH) and increases the expression of estrogen receptor beta (*Esr2*) in the paraventricular nucleus of the hypothalamus (PVH) (*Patisaul et al., 2003*; *Aquino et al., 2016*; *Sá et al., 2016*).

We hypothesized that tamoxifen alters estrogen receptor signaling in the hypothalamus-POA to mediate key negative side effects of tamoxifen therapy. To test this hypothesis, we modeled tamoxifen treatment in mice with a 28-day treatment course based on human dosage (*Slee et al., 1988*)

and asked if mice experience physiological effects similar to humans. We measured movement, bone density, and the temperature of the body core, tail skin, and thermogenic brown adipose tissue (BAT). Profiling genome-wide expression changes of individual cells in the hypothalamus-POA using Drop-seq, a droplet-based single-cell RNA sequencing technology, revealed transcriptional changes induced by tamoxifen in multiple cell types. Finally, we show that ERα expression in the hypothalamus-POA is necessary for the tamoxifen-induced chances in gene expression in the hypothalamus-POA and the effects on thermoregulation, bone density, and movement. Together, these findings suggest that tamoxifen treatment modulates ERα signaling in the central nervous system to alter fundamental aspects of physiology and health. Dissecting central versus peripheral effects and mechanisms of tamoxifen therapy is the first step toward identifying strategies to mitigate the adverse side effects of this life-saving treatment.

## Results

### Tamoxifen treatment alters thermoregulation

To ask if mice and humans experience similar physiological effects while undergoing tamoxifen treatment, we administered tamoxifen (0.1 mg/kg) or vehicle subcutaneously, daily, for 4 weeks (*Figure 1A*). In humans, hot flashes are characterized by frequent and sudden increases of heat dissipation from the face and other parts of the skin, often accompanied with perspiration and a decrease in core body temperature (*Stearns et al., 2002*). Similarly, tamoxifen-treated mice showed significantly lower core body temperature compared to controls, as indicated by 24 hr averages from the last week of treatment (Sidak's Post-hoc: 24 hr p=0.0076, see *Supplementary file 2* for detailed statistics). This difference is detected during the light phase when the animals are generally inactive (Light: p=0.0003) but not in the dark phase (Dark: p=0.1164) when movement can also influence core temperature (*Figure 1B and C*). In mice, heat dissipation occurs effectively via vasodilation in the tail (*Gordon, 1993*). As tail skin temperature is highly dependent on core and ambient temperature, heat loss is often expressed as the heat loss index (HLI): HLI = (Tskin − Tambient)/ (Tcore − Tambient) (*Romanovsky et al., 2002*). We observed a higher HLI in mice treated with tamoxifen compared to controls (*Figures 1D and E* and 24 hr: p=0.005). Again, this effect was detected in the light phase but not in the dark phase (*Figure 1E*, Light: p=0.0088, Dark: p=0.061). Tail skin temperature also was significantly higher in tamoxifen-treated mice compared to controls during light phase (*Figure 1—figure supplement 1A and B*, Light: p=0.0231, Dark: p=0.0968). In addition, mice treated with tamoxifen exhibited lower temperature above the intrascapular region directly apposed to BAT depots (*Figure 1F and G*, t = 3.255, df = 8, p=0.0116), suggesting reduced heat production from BAT following tamoxifen treatment. Accordingly, postmortem quantitative (q) PCR analysis of BAT revealed lower expression of genes associated with thermogenesis, uncoupling protein 1 (*Ucp1*) (t = 2.738, df = 16, p=0.0146) and adrenergic receptor beta 3 (*Adrb3*) (t = 2.732, df = 16, p=0.0148) (*Figure 1—figure supplement 1C*), suggesting suppressed BAT thermogenesis and sympathetic tone following tamoxifen treatment (*Cannon and Nedergaard, 2004*). Together, these results indicate that tamoxifen treatment shifts mouse temperature balance toward increased heat dissipation and decreased heat production, consistent with the observations in humans experiencing hot flashes.

### Tamoxifen treatment increases bone density and decreases movement

Tamoxifen has been shown to affect bone density in rodent and human studies (*Powles et al., 1996*; *Perry et al., 2005*). Here, micro-computed tomography (microCT) scans revealed that tamoxifen treatment is associated with greater bone volume ratios in both metaphysis (t = 7.322, df = 10, p<0.0001) and diaphysis (t = 3.557, df = 10, p=0.0052) of the femur (*Figure 2A–C*). Tamoxifen treatment was also associated with greater number (t = 7.539, df = 10, p<0.0001) and thickness (t = 5.303, df = 10, p=0.0003), and less separation of trabecular bones (t = 7.331, df = 10, p<0.0001) in metaphysis (*Figure 2B*). Bone mineral density (t = 0.09262, df = 10, p=0.928) and bone area (t = 1.368, df = 10, p=0.2013) in cortical bones were not significantly affected by tamoxifen treatment (*Figure 2C*); however, tamoxifen treatment was associated with significantly greater cortical thickness (t = 2.451, df = 10, p=0.0342) (*Figure 2A and C*). Together, these data indicate that chronic tamoxifen treatment increases bone mass, consistent with previous studies showing

increased bone formation and bone mass in mice following tamoxifen administration (*Perry et al., 2005*; *Starnes et al., 2007*). In addition, tamoxifen resulted in a moderate decrease in 24 hr movement (t = 3.296, df = 42, p=0.006) (*Figure 2D and E*). Tamoxifen-treated mice did not show significant changes in body weight compared to control mice, over 28 days of treatment (treatment: p=0.5468) (*Figure 2F and G*).

## Tamoxifen administration affects gene expression in all hypothalamus-POA cell types

To ask how tamoxifen affects gene expression in different cell types, we collected hypothalamus-POA (*Figure 3—figure supplement 1A*) after 28 daily injections of tamoxifen or vehicle and analyzed individual cells by drop-Seq (*Macosko et al., 2015*). A total of 29,807 cells (8220 from n = 3 vehicle-treated mice, 21,587 from n = 5 tamoxifen-treated mice) clustered into nine distinct cell types, annotated based on high expression of previously characterized cell-type markers (*Moffitt et al., 2018*; *Saunders et al., 2018*): astrocytes, oligodendrocytes, neurons, endothelial cells, microglia, polydendrocytes, ependymal cells, mural cells, and fibroblasts (UMAP: *Figure 3A*, tSNE: *Figure 3—figure supplement 1B*, cluster defining marker expression: *Figure 3—figure supplement 1D*). Because tamoxifen modulates estrogen receptor signaling (*de Médina et al., 2004*; *Jordan, 2007*), we examined cell-type specific expression of three estrogen receptor transcripts: the nuclear receptors *Esr1* and *Esr2,* as well as the G-protein-coupled estrogen receptor 1 (*Gper1*, formerly *Gpr30*) (*Revankar et al., 2005*). While neither *Esr1*, *Esr2*, nor the *Esr1* target gene *Pgr* showed strong enrichment in any particular cell type (*Figure 3B*), *Gper1* transcripts were predominantly found in Mural cells (LogFC: 1.39 compared to all other cell types, adj. p<1e-128). These results are in accordance with the reported *Esr1* expression pattern in the hypothalamus-POA (*Shughrue et al., 1997*) and *Gper1* in vascular smooth muscle of the rodent brain (*Isensee et al., 2009*). The graphical clustering methods, UMAP and tSNE, did not demonstrate clear separation between tamoxifen and vehicle treated cells (*Figure 3C*, *Figure 3—figure supplement 1C*), indicating that the effect of tamoxifen at clinically relevant doses may be modest with respect to global transcriptional signatures.

Among all cell types, neurons and ependymal cells were most sensitive to tamoxifen administration, as indicated by the number of significantly induced and repressed genes (*Figure 3D*). To account for the effect of cluster size on statistical power, we also calculated the number of significant differentially expressed genes per 1 k cells in each cluster. Again, neurons and ependymal cells have the highest numbers of differentially expressed genes when expressed as total or as a fraction of the number of cells sampled (*Figure 3—figure supplement 1E*), suggesting the strongest tamoxifen responsiveness. Although mural cells show significant enrichment of *Gper1*, tamoxifen administration was associated with relatively few significant gene expression changes in these cells (*Figure 3D*), indicating that *Gper1* does not likely mediate tamoxifen-induced gene expression in the hypothalamus-POA. It is possible, however, that tamoxifen affects cellular state in a non-transcriptional manner, via *Gper1*. Gene set enrichment analysis (GSEA) using a gene set for estrogen response (GO:0043627) further identifies neurons and ependymal cells as markedly tamoxifen responsive: tamoxifen administration induced a significant downregulation of genes involved in estrogen response in neurons, endothelial cells, and ependymal cells (*Figure 3E*, *Supplementary file 1a*). Individual gene expression changes that may be of interest can be queried at https://correalab.shinyapps.io/tamoxifenshiny/.

GSEA with a collection of 40 hallmark pathways and brain-focused gene ontology gene sets (*Supplementary file 1b*) revealed that all cell types except for mural cells showed significant enrichment or depletion of genes in at least one pathway (*Figure 3F*). Significantly enriched pathways (*Supplementary file 1c*) show both overlapping and distinct effects by cell type. For example, tamoxifen treatment decreased the expression of genes annotated as targets of the proto-oncogene, *Myc*, in astrocytes, oligodendrocytes, neurons, endothelial cells, microglia, polydendrocytes, and ependymal cells. Other cell-type specific pathways were also enriched or depleted, including establishment of the endothelial barrier in endothelial cells, neuropeptide signaling in neurons, E2F targets in astrocytes, and fatty acid metabolism in ependymal cells. Together these findings suggest that tamoxifen treatment has widespread effects on the hypothalamus-POA, altering transcriptional programs and cell function in both general and cell-type specific ways.

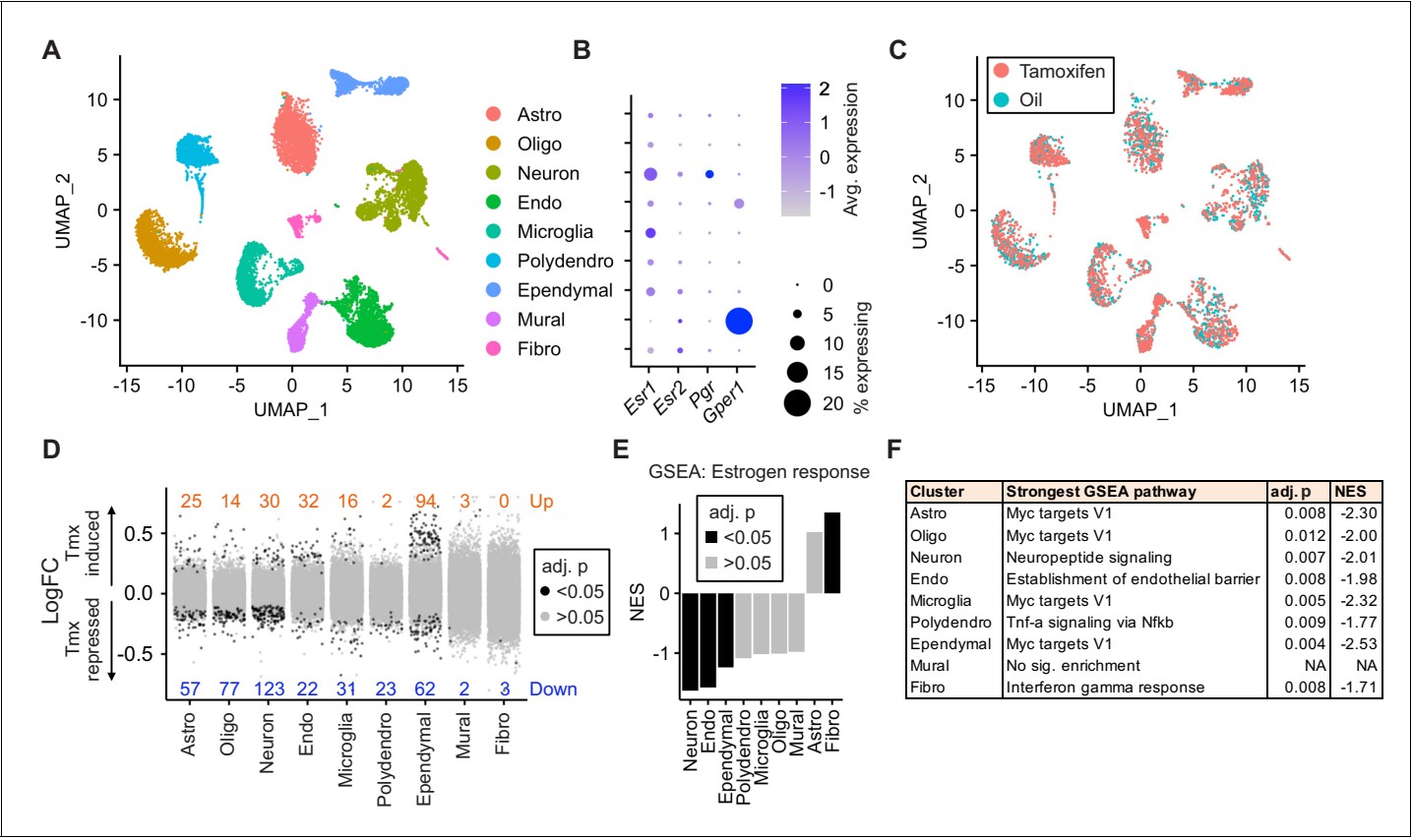

**Figure 3.** Tamoxifen treatment induces gene expression changes in various cell types of the hypothalamus and preoptic area (hypothalamus-POA). (**A**) UMAP showing clustering of the major cell types of the hypothalamus-POA based on single-cell transcriptomics. Each colored dot is a cell, with different colors representing different cell types listed to the right. (**B**) Dot plot showing expression of estrogen and progesterone receptors in cell types identified in (**A**). (**C**) UMAP comparing cells derived from mice receiving tamoxifen (pink) or oil injections (cyan). (**D**) Collapsed volcano plots showing differential gene expression, represented as log base 2 of fold change (LogFC), induced by daily tamoxifen treatment in various cell types identified. Up/down numbers refer to total number of significantly (Bonferroni adj. p<0.05) up- and downregulated genes. (**E**) Gene set enrichment analysis (GSEA) to find tamoxifen-induced signatures of estrogen response in hypothalamus-POA cell types. (**F**) Most strongly enriched or depleted pathways in GSEA comparing control to tamoxifen-treated cells. All analyses were done from cells harvested from female mice injected daily with oil (n = 3) or tamoxifen (n = 5) over 28 days. (**E–F**) NES: Normalized enrichment score, adj. p: Benjamini-Hochberg adjusted p-value.

The online version of this article includes the following figure supplement(s) for figure 3:

**Figure supplement 1.** Tamoxifen treatment induces gene expression changes in various cell types of the hypothalamus and preoptic area (hypothalamus-POA).

## Tamoxifen treatment causes gene expression changes in neuronal subtypes of the hypothalamus-POA

Given the transcriptional effects of tamoxifen treatment observed in neurons and the heterogeneity of estrogen-sensitive neurons within the hypothalamus-POA, we next examined the effect of tamoxifen treatment within individual neuronal clusters. Neuronal sub-clustering resulted in 25 apparent neuronal subtypes (*Figure 4A*). The majority of clusters were marked by expression of genes previously demonstrated to delineate neuronal types within the hypothalamus (*Figure 4B*; *Chen et al., 2017*; *Romanov et al., 2017*; *Kim et al., 2019*; *van Veen et al., 2020*). Markers of neuronal subtypes include neuropeptides, transcription factors, and cellular signaling messengers. *Esr1* expression was observed in various neuronal subclusters (*Figure 4B*), but was significantly enriched in wild-type neurons expressing the neuropeptide precursor, *Tac2* ($\log_2$FC = 1.52, adj. p=7.1e-36) and neurons expressing the neuropeptide precursor *Gal* ($\log_2$FC = 0.50, adj. p=1.6e-30) compared to all other neuronal subclusters. The ERα target gene *Pgr* also was significantly enriched in wild-type neurons expressing *Tac2* ($\log_2$FC = 0.49, adj. p=2.3e-4) compared to all other neuronal subclusters. GSEA demonstrates that overall, tamoxifen administration is associated with a significant down-

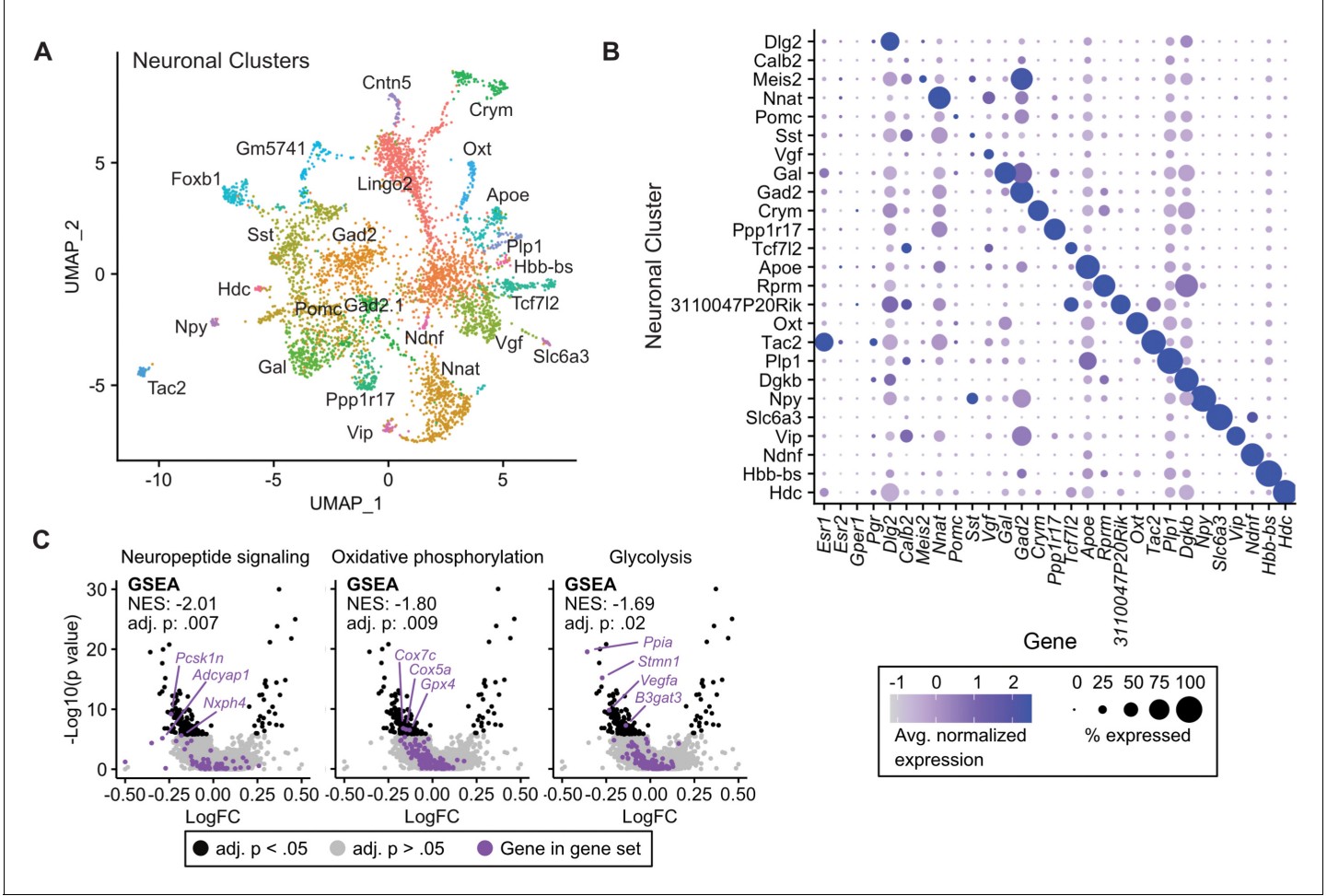

**Figure 4.** Tamoxifen-induced gene expression changes in neurons of the hypothalamus and preoptic area (hypothalamus-POA). (A) UMAP showing clustering of the neuronal subtypes of the hypothalamus-POA based on single-cell transcriptomics, overlayed with identity named for top expressed cluster defining marker gene. (B) Dot plot showing expression of neuronal cluster defining markers, *Esr1, Esr2, Gper1, and Pgr*. (C) Volcano plots of tamoxifen-induced or repressed differentially expressed genes (DEGs) overlayed with gene sets (GS) involved in neuropeptide signaling, oxidative phosphorylation, or glycolysis. Analyses done from wild-type female mice injected daily with oil (n = 3) or tamoxifen (n = 5) over 28 days. NES: Normalized enrichment score, GSEA adj. p: Benjamini-Hochberg adjusted p-value, DEG adj. p: Bonferroni adjusted p-value.

The online version of this article includes the following figure supplement(s) for figure 4:

**Figure supplement 1.** Tamoxifen-induced gene expression changes in hypothalamus and preoptic area (hypothalamus-POA) neurons.

regulation of genes involved in neuropeptide signaling, a critical aspect of neuronal function, as well as the two principal components fueling the sizeable energy demands of neuronal metabolism: oxidative phosphorylation and glycolysis (*Yellen, 2018*; *Figure 4C*). Notably, it has previously been shown that estrogens can directly regulate these metabolic pathways in the brain (*Brinton, 2008*; *Brinton, 2009*). When divided into subtypes, few statistically significant gene expression changes caused by tamoxifen were observed, likely due to loss of statistical power resulting from relatively few cells in each individual cluster (*Figure 4—figure supplement 1A*). To gain a preliminary view at the effect of tamoxifen on individual neuronal subtypes, we performed GSEA on all clusters; 14 of 25 neuronal types showed enrichment of at least one pathway (*Supplementary file 1d*). Neurons marked by expression of *Tac2* showed significant enrichment or depletion of the most pathways (*Figure 4—figure supplement 1B*). Interestingly, examining individual pathways shows differential responses by neuronal subtype. For example, genes involved in oxidative phosphorylation appear downregulated by tamoxifen in neurons expressing *Foxb1*, but the same gene set is upregulated in neurons expressing *Tac2* (*Figure 4—figure supplement 1B*). These cell-type specific effects

highlight the additional insights that can be revealed by scRNA-seq compared to overall averages determined by bulk tissue sequencing. Additionally, these results are compatible with the previous findings that tamoxifen can have different effects in different regions of the brain (*Wilson et al., 2003*; *Chen et al., 2014*; *Gibbs et al., 2014*; *Aquino et al., 2016*; *Sá et al., 2016*) and consistent with cell-type specific effects of tamoxifen within neurons of the hypothalamus-POA.

## *Esr1* conditional knockout ablates gene expression responses to tamoxifen

As *Esr1* is a known target of tamoxifen that is expressed in the hypothalamus-POA (*Figure 3B*), we sought to determine the impact of *Esr1* knockout on gene expression responses to tamoxifen administration. The NK2 homeobox transcription factor, *Nkx2-1*, is highly enriched in the medial basal hypothalamus and *Esr1^{F/F}*; *Nkx2-1^{Cre}* (*Esr1* cKO) mice display a selective loss of ERα immunoreactivity in the hypothalamus-POA (*Correa et al., 2015*; *Herber et al., 2019*). Here, single-cell RNA expression analysis was unable to detect a significant decrease of *Esr1* transcripts in cells of the hypothalamus-POA in *Esr1* cKO mice (*Figure 5—figure supplement 1A*), though both *Nkx2-1* and *Esr1* transcripts are present in various cell types of the hypothalamus-POA (*Figure 5—figure supplement 1B*), suggesting that the Cre and floxed alleles might be co-expressed within certain cell types. Indeed, 104 cells were found to express both *Esr1* and *Nkx2-1*. Of these 104, 79 were neurons, 19 were ependymal cells, 3 were astrocytes, 2 were microglia, and 1 was an endothelial cell. Most importantly, ERα immunoreactivity was clearly ablated in several areas of the hypothalamus-POA of *Esr1* cKO mice, including the medial preoptic area (MPA), ventrolateral subdivision of the ventromedial nucleus of the hypothalamus (VMHvl), and arcuate nucleus of the hypothalamus (ARC) (*Figure 5A*). Notably, the MPA, ARC, and VMHvl show highly enriched ERα expression in wild-type mice (*Merchenthaler et al., 2004*), and these regions mediate the effects of estrogens on body temperature (*Xu et al., 2011*; *Mauvais-Jarvis et al., 2013*; *Martínez de Morentin et al., 2014*), physical activity (*Correa et al., 2015*; *Krause et al., 2019*; *van Veen et al., 2020*), and bone regulation (*Farman et al., 2016*; *Herber et al., 2019*).

To test the effect of tamoxifen administration on gene expression in mice lacking ERα in the hypothalamus-POA, *Esr1* cKO mice received the same daily tamoxifen or vehicle injection regimen concurrently with wild-type mice (*Figure 1A*). UMAP Clustering of all four treatment conditions, vehicle or tamoxifen in wild-type or *Esr1* cKO animals, did not reveal clear separation in global transcriptional signature between any treatment group when considering all cell types (*Figure 5—figure supplement 1C*) or re-clustered neuronal subtypes (*Figure 5—figure supplement 1D*). *Esr1* cKO animals showed proportionally fewer astrocytes and more oligodendrocytes than wild-type animals (*Figure 5—figure supplement 1E*), though all other cell types were recovered at similar proportions. Tamoxifen treatment was not associated with any significant differences in recovered cell type proportions (*Figure 5—figure supplement 1E*).

Interestingly, tamoxifen treatment led to significant transcriptional changes in the hypothalamus-POA of *Esr1* cKO mice (*Figure 5B*). Similar to wild-type, *Esr1* cKO neurons and ependymal cells were the cell types most sensitive to tamoxifen administration compared to all other cell types, as indicated by the number of significantly induced and repressed genes (*Figure 5B*). To look closer at the specific effects of tamoxifen on gene regulation, and to ask if they depend on expression of *Esr1*, we examined those gene expression changes that were significant in wild-type cells and asked how those same genes were regulated in equivalent *Esr1* cKO cells. For this gene-by-gene comparison, there was a strong negative Spearman correlation observed in most cell types (*Figure 5C*). This negative correlation implies that the majority of significant tamoxifen-induced gene expression changes that were observed in cell types of wild-type female mice were ablated or regulated in the opposite direction in *Esr1* cKO animals.

In wild-type neurons, the majority of tamoxifen-regulated genes were repressed by tamoxifen (*Figure 3D*). In *Esr1* cKO neurons, most of these same genes are no longer repressed by tamoxifen (*Figure 5D*, Key: *Figure 5—figure supplement 2A*). In wild-type ependymal cells, the majority of regulated genes were induced by tamoxifen (*Figure 3D*) and the majority of these same genes were suppressed by tamoxifen in *Esr1* cKO ependymocytes (*Figure 5D*). Despite identifying fewer differentially expressed genes, the other cell types showed similar trends (*Figure 5—figure supplement 2B*). These data suggest that tamoxifen has generally suppressive effects on neuronal transcription

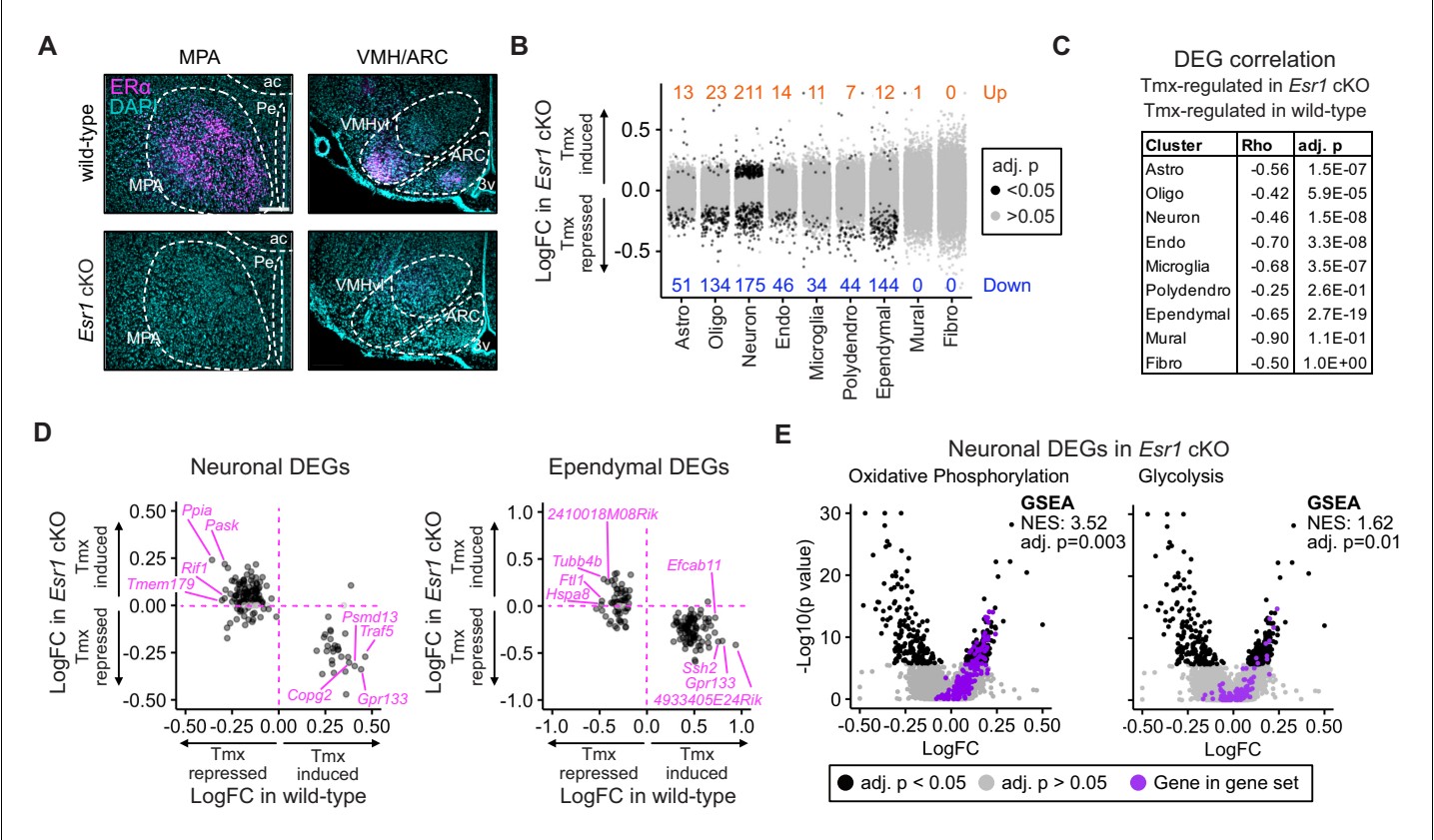

**Figure 5.** *Esr1* conditional knockout reverses hypothalamus and preoptic area (hypothalamus-POA) responses to tamoxifen. (A) Immunoreactive staining of estrogen receptor alpha (ERα) in the medial preoptic area (MPA), ventromedial nucleus of the hypothalamus (VMH), and the arcuate nucleus of the hypothalamus (ARC) of ERα knockout and wild-type female mice. Scale bar: 200 um. (B) Differentially expressed genes (DEGs) induced by daily tamoxifen treatment in cell types of the *Esr1^{f/f};Nkx2-1^{Cre}* (*Esr1* cKO) hypothalamus-POA. Up/down numbers refer to total number of significantly (Bonferroni adj. p<0.05) up- and downregulated genes. (C) Table of correlations showing how tamoxifen-induced gene expression changes in wild-type cells correlate with tamoxifen-induced gene expression changes in *Esr1* cKO cells. Rho: Spearman correlation coefficient, adj. p: Benjamini-Hochberg adjusted p-value. (D) Gene-by-gene comparison of how tamoxifen treatment affects expression in wild-type and *Esr1* cKO neurons and ependymal cells. Named genes are a subset of genes discordantly regulated by tamoxifen in wild-type and *Esr1* cKO cells. A complete list of genes shown here is in ***Supplementary file 1i***. (E) Volcano plots of all tamoxifen-induced or repressed DEGs (black) overlaid with gene sets (purple) involved in oxidative phosphorylation or glycolysis. NES: Normalized enrichment score, gene set enrichment analysis (GSEA) adj. p: Benjamini-Hochberg adjusted p-value, DEG adj. p: Bonferroni adjusted p-value. (B–E) Data from n = 4 oil treated *Esr1* cKO and n = 4 tamoxifen-treated *Esr1* cKO, n = 3 oil treated wild-type and n = 5 tamoxifen-treated wild-type mice.

The online version of this article includes the following figure supplement(s) for figure 5:

**Figure supplement 1.** Cell cluster analyses across all four treatment groups.

**Figure supplement 2.** Comparison of gene expression changes induced by tamoxifen in wild-type and *Esr1* cKO mice.

and generally inductive effects on ependymal cells; further, it suggests that these effects are largely dependent on expression of *Esr1*.

To determine if the overall opposite effects of tamoxifen in *Esr1* cKO compared to wild-type animals was associated with corresponding changes in functional pathways, we performed GSEA with all cell-type clusters as before. GSEA for estrogen responsive genes in all cell types showed that tamoxifen downregulated estrogen responsive genes in multiple cell types of the wild-type hypothalamus-POA but did not significantly affect the same gene set in cells of the *Esr1* cKO hypothalamus-POA, even when using a permissive cutoff for significance (***Figure 5—figure supplement 2C***, ***Supplementary file 1a and 1f***). GSEA in individual wild-type neuronal subclusters showed that tamoxifen downregulated estrogen responsive genes in various neuronal subtypes. Tamoxifen also downregulated estrogen responsive genes in some *Esr1* cKO neuronal subclusters (***Figure 5—figure supplement 2D***, ***Supplementary file 1g and 1h***), but this effect was attenuated compared to wild-

type. Many other functional pathways were altered in opposite directions in *Esr1* cKO cells compared to wild-type cells following tamoxifen treatment (complete results: *Supplementary file 1c and 1e*). This pattern included *Myc* targets in endothelial cells, ependymal cells, neurons, oligodendrocytes, and polydendrocytes. Importantly, we also found a significant change of metabolic gene expression in neurons, as both oxidative phosphorylation and glycolysis were upregulated by tamoxifen in *Esr1* cKO neurons (*Figure 5E*) but downregulated in wild-type neurons (*Figure 4C*). This trend of opposing effects of tamoxifen was not universal, as *Tnf-a* signaling was significantly downregulated by tamoxifen in both *Esr1* cKO and wild-type endothelial cells. Finally, many pathways, including *mTorc1* signaling in astrocytes, establishment of the endothelial barrier in endothelial cells, and neuropeptide signaling in neurons, were significantly affected by tamoxifen in wild-type cells, but not affected in their *Esr1* cKO counterparts (*Supplementary file 1c and 1e*). Together these data demonstrate that the majority of tamoxifen-induced changes in gene expression in the hypothalamus-POA are dependent upon expression of ERα in the *Nkx2-1* lineage. This includes many functional pathways (e.g., neuronal activity and estrogen responsive genes) in many cell types and neuronal subtypes.

### The homeostatic effects of tamoxifen are dependent on ERα

Given the known roles of ERα in mediating many aspects of physiology, as well as the effect of ERα conditional knockout on the transcriptomic effects of tamoxifen, we asked if loss of ERα would also affect physiological responses to tamoxifen in mice. We therefore measured body temperature, movement, and bone density in *Esr1* cKO mice treated equivalently and concurrently with wild-type animals (*Figure 1A*). Interestingly, tamoxifen injection was not associated with any difference in core body temperature in *Esr1* cKO mice compared to vehicle-treated controls (treatment: $F_{(1, 14)}$ =0.1395, p=0.7144) (*Figure 6A–B*). Specifically, we did not detect any significant differences in heat dissipation or production when comparing *Esr1* cKO mice treated with tamoxifen vs. vehicle, as indicated by HLI (treatment: $F_{(1, 13)}$=3.109, p=0.1013) (*Figure 6C–D*) and BAT temperature (t = 0.2640, df = 4, p=0.8048) (*Figure 6E–F*). In contrast to wild-type mice, there was no significant difference in movement between tamoxifen and vehicle treatment in *Esr1* cKO mice (treatment: $F_{(1, 14)}$=2.638, p=0.1266) (*Figure 6G–H*).

Knockout of ERα in the hypothalamus-POA largely blocked the effect of tamoxifen on bone physiology as well. No difference was observed in bone volume fraction (t = 0.8340, df = 10, p=0.4237), thickness (t = 0.2668, df = 10, p=0.7951), or separation of trabeculae (t = 1.924, df = 10, p=0.0832) when comparing tamoxifen to vehicle-treated animals. In contrast to wild-type mice, we observed a tamoxifen-induced reduction in trabecular number (t = 3.349, df = 10, p=0.0074) (*Figure 6I–J*). There was no significant difference in bone volume ratio (t = 0.5963, df = 10, p=0.5643) or cortical thickness (t = 0.9720, df = 10, p=0.3540) in the cortical bones of mice treated with tamoxifen compared to mice treated with vehicle (*Figure 6I and K*). Taken together, these results suggest that the dysregulation of core temperature, reduced movement, and changed bone physiology associated with tamoxifen treatment in wild-type mice are largely dependent upon the expression of ERα in the *Nkx2-1* lineage.

## Discussion

Tamoxifen is a selective estrogen receptor modulator that exerts antiproliferative effects on cancer cells through ERα (*Davies et al., 2011*), but can exert agonistic or antagonistic effects on the different estrogen receptors depending on cell type and context (*Watanabe et al., 1997*; *Mo et al., 2013*). The hypothalamus-POA is rich in estrogen receptor expression, making it a prime candidate as a tamoxifen target (*Merchenthaler et al., 2004*; *Gofflot et al., 2007*; *Matsuda et al., 2013*; *Saito et al., 2016*). Here, we take the first steps to define the cell types and cellular mechanisms that mediate some of the undesirable side effects experienced by people undergoing tamoxifen therapy. We demonstrate the utility of mice as a good model to study many of tamoxifen's physiological effects; tamoxifen administration in mice increases heat dissipation, decreases movement, and increases bone density. These effects are similar to those experienced by people receiving tamoxifen therapy, who experience hot flashes, lethargy, and changes in bone density, among other symptoms (*Love et al., 1991*; *Powles et al., 1996*; *Fisher et al., 1998*; *Loprinzi et al., 2000*; *Haghighat et al., 2003*; *Howell et al., 2005*; *Kligman and Younus, 2010*; *Francis et al., 2015*).

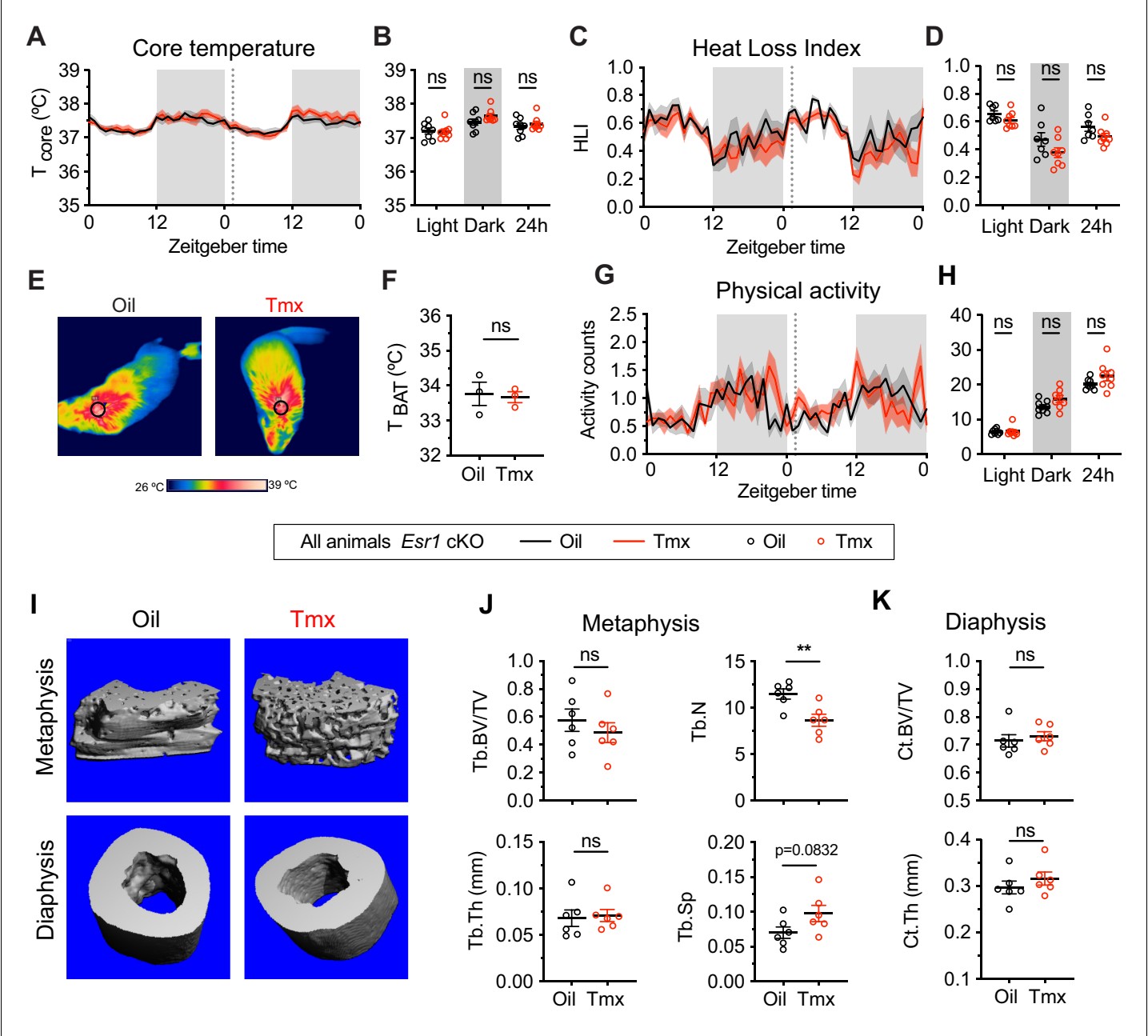

**Figure 6.** The homeostatic effects of tamoxifen are dependent on estrogen receptor alpha (ERα) in the hypothalamus and preoptic area (hypothalamus-POA). (A) Hourly average of core body temperature in *Esr1* cKO female mice over 2 days before and 3 weeks after injections (dotted line). (B) Average core body temperature from mice shown in panel (A) highlighting per animal averages in light (7:00 to 19:00), dark (19:00 to 7:00), and total 24 hr periods (n = 8, treatment: $F_{1,14}$=1.395, p=0.7144). (C) Heat loss index (HLI) calculated from continuous measurements of core and tail temperature. (D) Average HLI from mice shown in panel (C) (n = 7–8, treatment: $F_{1,13}$=3.109, p=0.1013). (E) Thermographic images of interscapular skin above brown adipose tissues (BAT) depots in *Esr1* cKO female mice injected with either oil or tamoxifen. (F) Quantification of temperature of skin above intrascapular BAT depots (n = 3, $t_4$ = 0.264, p=0.8048). (G) Hourly average of movement over 2 days before and after injections (dotted line). (H) Average movement from mice shown in panel (H) (n = 8, treatment: $F_{1,14}$=2.638, p=0.1266). (I) Representative micro-computed tomography (microCT) images showing bone density of the distal metaphysis and midshaft diaphysis of femurs from tamoxifen or oil treated *Esr1* cKO female mice. (J) Trabecular bone volume fraction (BV/TV, $t_{10}$ = 0.8340, p=0.4237), trabecular numbers (Tb.N, $t_{10}$ = 3.349, p=0.0074), trabecular thickness (Tb.Th, $t_{10}$ = 0.2668, p=0.7951), and trabecular separation (Tb.Sp, $t_{10}$ = 1.924, p=0.0832) in distal metaphysis of femurs (n = 6). (K) Bone volume fraction (Ct.BV/TV, $t_{10}$ = 0.5963, p=0.5643) and cortical thickness (Ct.Th, $t_{10}$ = 0.9720, p=0.354) in diaphysis of femurs (n = 6). Line shading and error bars represent standard error of the mean (SEM). ns, non-significant; **, p<0.01 for Sidak's multiple comparisons test (B, D, and H) or two-tailed t-test (F, J, and K). The online version of this article includes the following source data for figure 6:

**Source data 1.** Source data for *Figure 6*, panel a.

## eLife Research article

Cancer Biology | Neuroscience

**Source data 2.** Source data for *Figure 6*, panel b.
**Source data 3.** Source data for *Figure 6*, panel c.
**Source data 4.** Source data for *Figure 6*, panel d.
**Source data 5.** Source data for *Figure 6*, panel f.
**Source data 6.** Source data for *Figure 6*, panel g.
**Source data 7.** Source data for *Figure 6*, panel h.

Using single-cell RNA sequencing, we find that tamoxifen alters transcriptomes in the hypothalamus-POA, consistent with the hypothesis that tamoxifen treatment alters central nervous system function (*Eberling et al., 2004*; *Gibbs et al., 2014*; *Denk et al., 2015*). Indeed, tamoxifen and its metabolites are detectable and up to 46-fold higher in the brain than in serum of breast cancer patients (*Lien et al., 1991*). We report that tamoxifen treatment affects gene expression in many cell types but most strongly in neurons and ependymal cells. These findings are consistent with previous reports of tamoxifen-induced gene expression changes within neurons (*López et al., 2006*; *Aquino et al., 2016*; *Sá et al., 2016*), and tamoxifen-induced stress in neurons and ependymal cells (*Denk et al., 2015*). Although much higher doses, such as those used to activate tamoxifen-inducible mouse models (25–100 mg/kg), can inhibit neural progenitor cell proliferation during cortical patterning or induce adipogenesis and prolonged genetic effects (*Ye et al., 2015*; *Lee et al., 2020*), we do not detect any changes in the proportion of different cell types after 28 days of treatment with a clinically relevant dose (0.1 mg/kg) of tamoxifen.

Notably, conditional knockout of the *Esr1* gene encoding ERα did not render the brain insensitive to tamoxifen. Instead it ablated or reversed the direction of a significant number of the tamoxifen-induced gene expression changes that were observed in the wild-type hypothalamus-POA. Importantly, this reversal encompassed many functional pathways including those controlling metabolism and neuropeptide signaling in neurons. The many transcriptional changes induced by tamoxifen in *Esr1* cKO mice might be attributed to tamoxifen's effects on the other estrogen receptors or effects mediated by cells outside of the *Nkx2-1* lineage. Indeed, studies of breast cancer resistance suggest many other signaling pathways, for example, Pgr, androgen receptor, and GPER may be involved in the actions of tamoxifen on transcription (*Mo et al., 2013*; *Rondón-Lagos et al., 2016*). Pgr expression in the hypothalamus is regulated by both endogenous estrogens and tamoxifen (*Sá et al., 2016*) and we find that *Pgr* is most abundant in neurons. Although more enriched in mural cells, GPER has been shown to increase acetylcholine release in neurons of the hippocampus (*Gibbs et al., 2014*). Nevertheless, conditional knockout of ERα in the hypothalamus-POA blocked the majority of the physiological changes observed in wild-type mice by tamoxifen, indicating a pivotal role of ERα in mediating the homeostatic effects of tamoxifen. Together, these data indicate an indispensable role for ERα in the molecular and physiological response to tamoxifen treatment in mice. In addition, our data indicate that tamoxifen exerts potent effects on gene expression in a variety of neuronal subtypes, as well as in ependymal cells. It is very likely that the pleiotropic physiological effects of tamoxifen involve various estrogen responsive cells and brain regions. Determining which regions of the hypothalamus-POA respond directly to tamoxifen and which nuclei mediate the various effects of tamoxifen will require careful region and cell-type specific *Esr1* deletion studies.

Hot flashes are one of the most common complaints by people undergoing tamoxifen therapy or transitioning to menopause (*Stearns et al., 2002*; *Kligman and Younus, 2010*). Hot flashes are characterized by frequent, sudden increases of heat dissipation from the skin, often accompanied with sweats and transient decreases in core body temperature (*Stearns et al., 2002*). The precise mechanisms behind this change in thermoregulatory responses are unknown; however, several studies have suggested that a disruption in central sex hormone signaling is involved. The hypothalamus-POA is a central site for both sex hormone action and body temperature regulation and appears to play a role in the etiology of hot flashes. Specifically, activating neurons in the ARC that co-express kisspeptin, neurokinin B, and dynorphin (KNDy neurons, which also express ERα) or manipulating their downstream neurokinin B signaling in the POA induces hot flash-like symptoms in rodents and humans (*Dacks et al., 2011*; *Jayasena et al., 2015*; *Padilla et al., 2018*). We detected the highest *Esr1* expression levels in neurons marked by expression of the neurokinin B precursor gene, *Tac2*, suggesting that tamoxifen may interact with neurokinin B signaling through ERα in the ARC or the

POA. Indeed, *Esr1* cKO mice which showed loss of ERα in the ARC and POA (*Figure 5A*) did not exhibit tamoxifen-induced tail and core temperature changes. Together, these data suggest the hypothesis that tamoxifen induces hot flashes via estrogen-sensitive neural circuits involved in thermoregulation. However, it is important to acknowledge that the sustained increase in tail skin temperature observed here and in other animal models (*Kobayashi et al., 2000*; *Padilla et al., 2018*) does not fully reflect the episodic nature of hot flashes in humans. Additionally, there is evidence that other ERs can contribute to estrogen's effects on thermoregulation. For example, treating ovariectomized rats with selective ligands for ERβ can mimic the effects of estradiol treatment on tail skin temperature (*Opas et al., 2009*). Similarly, a Gq-coupled estrogen receptor ligand (STX) is able to decrease core body temperature similar to E2 in guinea pigs after ovariectomy (*Roepke et al., 2010*). Although the effects of estrogens on thermoregulation are probably multifaceted and complex, it is clear that ablating ERα is sufficient to abrogate the effects of tamoxifen on core body temperature and tail skin temperature in mice.

Animal studies indicate that estrogen signaling promotes movement (*Ogawa et al., 2003*; *Lightfoot, 2008*), although in humans, estrogen related activity changes remain controversial (see review *Bowen et al., 2011*). In female breast cancer patients, there is a significant association between ongoing tamoxifen usage and symptoms of fatigue (*Haghighat et al., 2003*). Our results demonstrate that ERα in the hypothalamus-POA mediates a suppressive effect of tamoxifen on movement. This is consistent with the results that ERα signaling in the hypothalamus-POA regulates physical activity in mice (*Ogawa et al., 2003*; *Musatov et al., 2007*; *Xu et al., 2011*; *Sano et al., 2013*; *Correa et al., 2015*; *van Veen et al., 2020*). Specifically, activation of ERα neurons in the *Nkx2-1* lineage promotes movement (*Correa et al., 2015*) and loss of ERα in the VMH reduces physical activity in mice (*Musatov et al., 2007*; *Correa et al., 2015*). In addition, the POA is rich in ERα expression and has been shown to be responsible for estrogen induced running wheel activity (*King, 1979*; *Fahrbach et al., 1985*; *Takeo and Sakuma, 1995*). Although ERβ also is expressed in the hypothalamus-POA, physical activity is primarily regulated by ERα signaling in mice (*Ogawa et al., 2003*). Therefore, tamoxifen may act as an ERα antagonist in those estrogen-sensitive regions to suppress physical activity in mice.

Similar to endogenous estrogens, tamoxifen has profound effects on bone remodeling. Human studies have demonstrated that tamoxifen is protective against bone mineral density loss after menopause but induces bone loss before menopause (*Love et al., 1992*; *Kristensen et al., 1994*; *Powles et al., 1996*; *Vehmanen et al., 2006*). In contrast, animal studies have generally shown protective effects of tamoxifen on bone, regardless of ovarian function (*Turner et al., 1988*; *Perry et al., 2005*; *Starnes et al., 2007*). Whether these discrepancies are due to fundamental differences between rodent and human bone biology remains to be determined. We found that tamoxifen greatly increased bone mass in wild-type mice, displaying an estrogen-like protective effect. Although this study did not evaluate bone formation and resorption respectively, a number of animal studies have indicated that tamoxifen can stimulate bone formation (*Perry et al., 2005*) and also suppress bone resorption (*Turner et al., 1988*; *Quaedackers et al., 2001*), resulting in a net bone accruement. We did not detect an effect of tamoxifen on bone volume in *Esr1* cKO mice and observed opposite changes on trabecular bone number compared to that in wild-type mice. These results suggest that at least some of the effects of tamoxifen on bone are mediated by ERα signaling within the *Nkx2-1* lineage. ERα signaling outside of the *Nkx2-1* lineage or other estrogen receptor subtypes may also contribute to the effects observed in wild-type mice (*Windahl et al., 2002*). Indeed, E2 replacement after ovariectomy can increase cortical bone dimensions in ERα and ERβ double KO mice (*Lindberg et al., 2002*) and a Gq-coupled estrogen receptor ligand is able to increase bone density in guinea pigs after ovariectomy (*Roepke et al., 2010*). However, depletion of ERα in the medial basal hypothalamus or specifically within the KNDy neurons results in an impressive increase in bone mass (*Farman et al., 2016*; *Herber et al., 2019*), suggesting that the strongest effects of estrogens on bone may be mediated by hypothalamic ERα. The potent effect of hypothalamic ERα on bone is consistent with our finding that conditional ERα ablation in the hypothalamus-POA can abrogate the effects of tamoxifen on bone volume. However, it is possible that we were unable to detect more subtle effects of tamoxifen on bone in *Esr1* cKO mice, as these could be masked by the dramatic alteration in bone metabolism observed in *Esr1* cKO mice.

Although the *Nkx2-1* lineage includes cells in the thyroid, pituitary, and lung, *Esr1* expression is not altered in these peripheral tissues of *Esr1* cKO mice (*Herber et al., 2019*), leaving the

hypothalamus-POA as the most likely mediator of the ERα-dependent physiological effects of tamoxifen reported here. Despite this, it is very likely that peripheral tissues are also affected by systemic tamoxifen administration, leading to other physiological effects that we did not examine. Indeed, there is evidence that estrogen suppression therapies, including tamoxifen or aromatase inhibitors, can lead to the dysregulation of energy balance and glucose homeostasis in humans and when modeled in rodents (*López et al., 2006*; *Lampert et al., 2013*; *Ceasrine et al., 2019*; *Scalzo et al., 2020*). Together, these studies reveal widespread effects of tamoxifen treatment on the hypothalamus-POA and implicate ERα signaling within the *Nkx2-1* lineage as a major mediator of the effects of tamoxifen on thermoregulation, movement, and bone homeostasis.

In summary, the rodent model of tamoxifen treatment provided here mimics several of the key side effects of endocrine therapy observed in humans. If translatable to humans, our findings suggest the hypothesis that tamoxifen treatment may alter thermoregulation, physical activity, and bone through ERα signaling in the brain. A recent survey conducted in the online breast cancer communities showed that women often experienced more of tamoxifen's side effects than men and one-third of patients did not feel that their side effects were taken seriously (*Berkowitz et al., 2021*). The insights provided here are a first step toward the development of alternative or improved treatments that mitigate or circumvent some of the undesirable side effects of tamoxifen therapy.

# Materials and methods

**Key resources table**

| Reagent type (species) or resource | Designation | Source or reference | Identifiers | Additional information |
|---|---|---|---|---|
| gene (*Mus musculus*) | *Esr1* | MGI | MGI:1352467 NCBI Gene: 13982 | |
| Strain, strain background (*Mus musculus*) | *Esr1*[tm1Sakh] *Esr1 floxed* mouse line; CD-1;129P2 mixed background | *Feng et al., 2007* | MGI:4459300 | Targeted conditional mutation allele. Females used for experiments |
| Strain, strain background (*Mus musculus*) | *Tg(Nkx2-1-cre)*[2Sand] *Nkx2-1Cre* mouse line CD-1;129P2 mixed background | *Xu et al., 2008* | MGI: 3773076 | BAC transgenic Cre driver. Females used for experiments |
| Antibody | anti-ERα (mouse monoclonal) | Santa Cruz | Cat# sc-8002, RRID:AB_627558 | IF (1:250) |
| Chemical compound, drug | Tamoxifen | Sigma-Aldrich | T5648 | |
| Software, algorithm | Seurat R package | *Butler et al., 2018* | version 3.2.0 | Single-cell RNA sequencing analyses |
| Software, algorithm | Prism | GraphPad | version 8 | Physiology data analyses |
| Software, algorithm | VitalView software | Starr Life Sciences | version 5.1 | Core temperature and movement data collection |
| Software, algorithm | Mercury Analysis Software | Star:ODDI | version 5.7 | Tail temperature data collection |
| Other | DAPI stain | Invitrogen | D1306, RRID:AB_2629482 | IF (1 μg/mL) |

## Mice

All mice were maintained under a 12:12 hr L/D schedule at room temperature (22–23°C) and provided with food and water ad libitum. Mice expressing the NK2 homeobox transcription factor 1 (*Nkx2-1*; also known as *Ttf1*) Cre driver transgene (Tg(*Nkx2-1-Cre*)[2Sand]), and the *Esr1* floxed allele (*Esr1*[tm1Sakh]) were maintained on a CD-1;129P2 mixed background. *Cre*-negative littermates were selected as controls. A total of 50 female mice were used. Mice were 8–10 weeks old at the start of

the experiments. All studies were carried out in accordance with the recommendations in the Guide for the Care and Use of Laboratory Animals of the National Institutes of Health. UCLA is accredited by the Association for Assessment and Accreditation of Laboratory Animal Care International (AAA-LAC) aThe UCLA Institutional Animal Care and Use Committee (IACUC) approved all animal procedures.

## Tamoxifen administration

Tamoxifen (Sigma-Aldrich, T5648) was dissolved first in ethanol and then diluted in corn oil at a final concentration of 100 µg/mL and 0.5% of ethanol. Accordingly, the vehicle was prepared in corn oil containing 5% ethanol. We performed daily subcutaneous injection of tamoxifen at a dose of 0.1 mg/kg or an equal volume of vehicle control for 4 weeks. We estimated that this dose models human tamoxifen exposure based on studies of tamoxifen in human serum, estimates of total blood volume in mice, and bioavailability of tamoxifen delivered via subcutaneous injection (*Slee et al., 1988*). Additionally, it is on the low end of what is used effectively in previous studies (*Wade and Heller, 1993*; *Perry et al., 2005*), which is desirable to minimize off-target effects.

## Telemetry recording

Mice were anaesthetized with isofluorane and received combinatorial analgesics (0.01 mg/mL bupre-norphine, 0.58 mg/mL carprofen) pre and post any surgeries. A G2 eMitter (Starr Life Sciences) was implanted to the abdominal cavity and attached to the inside body wall of a mouse. Mice were single-housed in cages placed on top of ER4000 Energizer/Receivers. Gross movement and core body temperature were measured every 5 min using VitalView software (Starr Life Sciences). These measurements were collected continuously for 3 days on and 3 days off for the duration of the 28-day period. Telemetry data show 24 hr averages from the two telemetry sessions before (baseline) and 3 weeks of tamoxifen or vehicle injections (treatment). Tail skin temperature was monitored every 5 min using a Nano-T temperature logger (Star-Oddi) that was attached to the ventral surface and 1 cm from the base of the tail in a 3D-printed polylactic acid collar modified from *Krajewski-Hall et al., 2018*. Tail skin data show 24 hr averages for the same time periods as the telemetry data.

## Thermal imaging

Infrared thermal images were captured using e60bx thermogenic camera (FLIR Systems) and analyzed using the FLIR Tools software. All images were obtained at a constant distance to subject in awake animals at light phase. BAT skin temperature was defined from the average temperature of a spherical area centered on the interscapular region.

## Single-cell dissociation and library preparation

To avoid circadian differences and batch effects, all four groups were performed in parallel at the same time across 4 days with n = 4 mice per day. Mice were euthanized 2 hr after the last administration of tamoxifen or vehicle. The brain was freshly collected into ice-cold HABG buffer (Hibernate A, B27, Glutamax, Fisher Scientific, Hampton, NH, USA) containing freshly prepared papain. A coronal section containing the hypothalamus-POA was quickly dissected along the boundary of rostral and caudal spherical grooves from the bottom of the brain. A third cut was made along the white fiber of anterior commissure to get a square block of tissue (*Figure 3—figure supplement 1A*). The tissue was then disassociated and prepared into a single-cell suspension at a concentration of 100 cells/µL in 0.01% BSA-PBS using a previously described protocol (*Liu et al., 2020*). Briefly, dissected tissues were incubated in a papain solution for 30 min at 30°C then washed with HABG. Using a siliconized 9-in Pasteur pipette with a fire-polished tip, the cells were triturated carefully to help dissociate the tissue. Next, to separate the cells and remove debris, the cell suspension was placed on top of a prepared OptiPrep density gradient (Sigma-Aldrich, St. Louis, MO, USA) then centrifuged at 800 g for 15 min at 22°C. After debris removal, the cell suspension containing the desired cell fractions was centrifuged for 3 min at 22°C at 200 g, and the supernatant was discarded. Finally, the cell pellet was re-suspended in 0.01% BSA (in PBS) and filtered through a 40 µm strainer (Fisher Scientific, Hampton, NH, USA). The cells were then counted and diluted to appropriate cell density.

Barcoded single cells, or STAMPs (single-cell transcriptomes attached to microparticles), and cDNA libraries were prepared as described previously (*Liu et al., 2020*) and in line with the online protocol v3.1 from McCarroll Lab (http://mccarrolllab.org/download/905/). Briefly, to generate STAMPS, the prepared single-cell suspensions, EvaGreen droplet generation oil (BIO-RAD, Hercules, CA, USA), and ChemGenes barcoded microparticles (ChemGenes, Wilmington, MA, USA) containing unique molecular identifiers (UMIs) and cell barcodes were co-flowed through a FlowJEM aquapel-treated Drop-seq microfluidic device (FlowJEM, Toronto, Canada) at recommended flow speeds (oil: 15,000 µL/h, cells: 4000 µL/h, and beads 4000 µL/h). After breakage of the droplets, the beads were washed and suspended in reverse transcriptase solution. Prior to PCR amplification, samples underwent an exonuclease I treatment. Next, the beads were washed, counted, and aliquoted into PCR tubes (6000 beads/tube) for PCR amplification (4+11 cycles). The cDNA quality was checked using a High Sensitivity chip on BioAnalyzer (Agilent, Santa Clara, CA, USA) then fragmented using Nextera DNA Library Preparation kit (Illumina, San Diego, CA, USA) with multiplex indices. The libraries were then purified, quantified, and sequenced on an Illumina HiSeq 4000 (Illumina, San Diego, CA, USA).

## Analysis of single-cell RNA-seq

Single-cell transcriptomic data were analyzed in R version 3.6.1 using the package 'Seurat' version 3.2.0 and custom Seurat helper package 'ratplots' version 0.1.0 written for this study. All custom functions and complete analysis scripts are available at https://github.com/jevanveen/; *Stuart et al., 2019*. Cells were filtered for quality with the following criteria: cells with >15% of reads coming from mitochondrial genes were excluded and cells with fewer than 200 or more than 4000 genes detected were excluded. Samples derived from the four experimental conditions—wild-type, vehicle treated; wild-type, tamoxifen treated; *Esr1 cKO*, vehicle treated; and *Esr1 cKO*, tamoxifen treated—were aligned based on the expression of 2000 conserved highly variable markers in each group. Cells were clustered based on transcriptome similarity, using a shared nearest neighbor algorithm (*Waltman and van Eck, 2013*) and displayed with both universal manifold approximation and projection (UMAP) and t-distributed stochastic neighbor embedding (tSNE). For each cell cluster, marker genes were determined by comparing expression in the given cluster against all other clusters using the smart local moving algorithm to iteratively group clusters together (*Blondel et al., 2008*). For all differentially expressed gene marker analyses, statistical significance testing was performed with the Seurat default Wilcoxon rank-sum-based test and Benjamini-Hochberg method for multiple-testing correction. GSEAs were performed in R using the package 'fgsea' version 1.14.0. GSEAs to find estrogen signatures were done using the gene ontology (GO) pathway 'GO_ESTRO-GEN_RESPONSE' taken from mSigDB version 6.2. For general pathway analyses, a custom set of pathways was assembled from mSigDB 'Hallmark' pathways version 6.2 and selected GO pathways (version 6.2) particularly relevant to neural development and function. The complete list of these pathways is given in *Supplementary file 1b*.

Cells annotated as neurons were extracted and re-clustered using the same steps used to cluster all cells, with the following differences: 500 highly variable genes were used to align neurons from different treatment groups. No additional quality filtering was performed. Neuronal subtypes were annotated based on top marker genes, defined as the highest LogFC compared to all other neuronal subtypes, while meeting the criterion of adjusted p-value < 0.05. GSEAs were performed on neuronal subtypes using the same gene sets and methods performed for general cell types. All graphs and tables in *Figures 3–5* were produced using the R packages 'ggplot2' version 3.3.2 and custom functions in 'ratplots' version 0.1.0.

## Micro-computed tomography

Left femoral bones from the hind legs were dissected, cleaned of any soft tissue, and frozen in PBS at −20℃ till further analysis. Samples were scanned in 70% ethanol using Scanco Medical µCT 50 specimen scanner with a voxel size of 10 mm, an X-ray tube potential of 55 kVp and X-ray intensity of 109 µA. Scanned regions included 2 mm region of the femur proximal to epiphyseal plate and 1 mm region of the femoral mid-diaphysis. For the analysis, a trabecular bone compartment of 1 mm length proximal to the epiphyseal plate was measured. Cortical parameters were assessed at the diaphysis in an adjacent 0.4 mm region of the femur. In both specimen and in vivo scanning, volumes

of interest were evaluated using Scanco evaluation software. Representative 3D images created using Scanco Medical mCT Ray v4.0 software.

## Immunohistochemistry

Mice were perfused transcardially with ice-cold PBS (pH = 7.4) followed by 4% paraformaldehyde (PFA) in PBS. Brains were post fixed in 4% PFA overnight, dehydration in 30% sucrose for 24 hr, embedded in optimal cutting temperature (OCT) compound, and stored in −80℃ before sectioning. Coronal sections were cut under cryostat (Vibratome) into eight equal series at 18 µm. ERα immunoreactivity was detected using hot based antigen retrieval immunohistochemistry protocol. Briefly, sections were first incubated in antigen retrieval buffer (25 mM Tris–HCl, 1 mM EDTA, and 0.05% SDS, pH 8.5) at 95℃ for 40 min. Sections were then blocked for 1 hr in 10% BSA and 2% normal goat serum (NGS) and incubated overnight at 4℃ with primary antibody (ERα, 1:250, sc-8002, Santa Cruz). Following 3 × 10 min washing in PBS, sections were incubated with fluorophore conjugated goat anti-mouse secondary antibody (Thermo Fisher Scientific) for 2 hr at room temperature. After washing, sections were incubated with DAPI, washed with PBS, and coverslipped with Fluoromount-G.

The images were taken by DM1000 LED fluorescent microscope (Leica). Cyan/magenta/yellow pseudo-colors were applied to all fluorescent images for color-friendly accessibility. Image processing was performed using the Leica Application Suite (Leica) and ImageJ (NIH).

## RNA isolation and real time PCR (qPCR)

Interscapular BAT was snap-frozen in liquid nitrogen and stored at −80℃ until analysis. Total RNA from BAT was isolated using Zymo RNA isolation kit (ZYMO Research) and RNA yield was determined using a NanoDrop D1000 (Thermo Fisher Scientific). cDNA synthesis was performed with equal RNA input using the Transcriptor First Strand cDNA synthesis kit (Roche Molecular Biochemicals). qPCR was performed using C1000 Touch Thermal Cycler (BioRed) and SYBR mix (Bioline, GmbH, Germany), using validated primer sets (*Zhang et al., 2020*) (*Ucp1*: F - CACGGGGACC TACAATGCTT and R – TAGGGGTCGTCCCTTTCCAA; *Adr3b* F - GGAAGCTTGCTTGATCCCCA and R - GCCGTTGCTTGTCTTTCTGG).

## Statistics

Data are represented as mean ± standard error of the mean (SEM). Data with normal distribution and similar variance were analyzed for statistical significance using two-tailed, unpaired Student's t-tests. Time course data were analyzed by repeated measures two-way ANOVA or mixed model followed by Sidak's multiple comparisons. Significance was defined at a level of α <0.05. Detailed statistics are listed in *Supplementary file 2*. Statistics were performed using GraphPad Prism eight and RStudio.

## Acknowledgements

This research was supported by V Scholar Awards (V2017-007 and DVP2020-005) from the V Foundation for Cancer Research to SMC, NIH R21CA249338 to SMC, two Pilot Awards from the Iris Cantor-UCLA Women's Health Center/UCLA National Center of Excellence in Women's Health and NIH National Center for Advancing Translational Science (NCATS) UCLA CTSI (UL1TR001881) to ZZ, JEV, and SMC. ZZ was supported by an American Heart Association Postdoctoral Fellowship (18POST33960457), and GD was supported by NIEHS NIH T32ES015457.

## Additional information

### Funding

| Funder | Grant reference number | Author |
| --- | --- | --- |
| National Cancer Institute | R21 CA249338 | Stephanie M Correa |
| V Foundation for Cancer Research | V Scholar Award (V2017-007) | Stephanie M Correa |

| National Center for Advancing Translational Sciences | Women's Health Pilot Project Grants in 2018 and 2019 | Zhi Zhang<br>J Edward van Veen<br>Stephanie M Correa |
| National Center for Advancing Translational Sciences | UCLA CTSI (UL1TR001881) | Zhi Zhang<br>J Edward van Veen<br>Stephanie M Correa |
| American Heart Association | 18POST33960457 | Zhi Zhang |
| NIEHS | T32ES015457 | Graciel Diamante |
| V Foundation for Cancer Research | DVP2020-005 | Stephanie M Correa |

The funders had no role in study design, data collection and interpretation, or the decision to submit the work for publication.

### Author contributions

Zhi Zhang, Conceptualization, Data curation, Formal analysis, Supervision, Funding acquisition, Validation, Investigation, Visualization, Methodology, Writing - original draft, Project administration; Jae Whan Park, Data curation, Formal analysis, Validation, Investigation, Methodology; In Sook Ahn, Graciel Diamante, Validation, Investigation, Writing - review and editing; Nilla Sivakumar, Data curation, Software, Formal analysis, Investigation, Writing - review and editing; Douglas Arneson, Conceptualization, Resources, Data curation, Software, Formal analysis, Supervision, Validation, Methodology, Writing - review and editing; Xia Yang, J Edward van Veen, Conceptualization, Resources, Data curation, Software, Formal analysis, Supervision, Funding acquisition, Validation, Investigation, Visualization, Methodology, Writing - original draft, Project administration, Writing - review and editing; Stephanie M Correa, Conceptualization, Resources, Formal analysis, Supervision, Funding acquisition, Validation, Investigation, Visualization, Methodology, Project administration, Writing - review and editing

### Author ORCIDs

Zhi Zhang https://orcid.org/0000-0002-3222-2877
J Edward van Veen https://orcid.org/0000-0003-1798-3210
Stephanie M Correa https://orcid.org/0000-0002-6221-689X

### Ethics

Animal experimentation: These studies were performed in strict accordance with the recommendation Guide for the Care and Use of Laboratory Animals of the National Institutes of Health. All of the animals were handled according to approved institutional animal care and use committee (IACUC) protocols (#2016-003B and #2016-004) at UCLA. The protocol was approved by the Animal Research Committee at UCLA. All surgery was performed under isoflurane anesthesia and every effort was made to minimize suffering.

### Decision letter and Author response

Decision letter https://doi.org/10.7554/eLife.63333.sa1
Author response https://doi.org/10.7554/eLife.63333.sa2

# Additional files

### Supplementary files

• Source code 1. R Code for scRNA-seq analyses.

• Supplementary file 1. Tables for single-cell data. (a) Gene set enrichment analysis (GSEA) of estrogen responsive genes in cells from tamoxifen-treated vs. vehicle-treated wild-type mice. (b) List of all gene sets queried in GSEAs for (c, d, and e). (c) Pathways significantly altered in cell clusters from tamoxifen-treated vs. vehicle-treated wild-type mice. (d) Pathways significantly altered in neuronal subtypes from tamoxifen-treated vs. vehicle treated wild-type mice. (e) Pathways significantly altered in cells from tamoxifen-treated vs. vehicle-treated *Esr1* cKO mice, as determined by GSEA. (f) GSEA

of estrogen responsive genes in cells from tamoxifen-treated vs. vehicle-treated *Esr1* cKO mice. (g) GSEA of estrogen responsive genes in neuronal clusters from tamoxifen-treated vs. vehicle-treated wild-type mice. (h) GSEA of estrogen responsive genes in neuronal clusters from tamoxifen-treated vs. vehicle-treated *Esr1* cKO mice. (i) Differentially expressed genes (DEGs) in neurons from tamoxifen-treated vs. vehicle-treated mice. (j) DEGs in ependymal cells from tamoxifen-treated vs. vehicle-treated mice.

- Supplementary file 2. Details of all statistical tests for physiology data. (a) Statistical details for *Figure 1*, panels c, e, and g. (b) Statistical details for *Figure 1—figure supplement 1*, panels b and c. (c) Statistical details for *Figure 2*, panels b and c. (d) Statistical details for *Figure 2*, panels e-g. (e) Statistical details for *Figure 6*, panels b-h. (f) Statistical details for *Figure 6*, panels j and k .(g) Statistical details for *Figure 5—figure supplement 1*, panel e.

- Transparent reporting form

## Data availability
Sequencing data have been deposited in GEO under accession number GSE158960.

The following dataset was generated:

| Author(s) | Year | Dataset title | Dataset URL | Database and Identifier |
|---|---|---|---|---|
| Zhang Z, Park JW, Ahn IS, Diamante G, Sivakumar N, Arneson DV, Yang X, Veen JE, Correa SM | 2021 | Hypothalamic estrogen receptor alpha mediates key side effects of tamoxifen therapy in mice | https://www.ncbi.nlm.nih.gov/geo/query/acc.cgi?acc=GSE158960 | NCBI Gene Expression Omnibus, GSE158960 |

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
