## [Decision Letter]

**Acceptance summary:**

Millions of women endure side effects associated with long-term tamoxifen therapy for prevention of breast cancer recurrence. Your work in the mouse demonstrating that estrogen receptors in specific regions of the brain mediate many of the peripheral effects of tamoxifen provides important insights into the mechanisms by which bone density, body temperature and movement are impacted this therapeutic.

**Decision letter after peer review:**

Thank you for submitting your article "Hypothalamic estrogen receptor alpha mediates key side effects of tamoxifen therapy in mice" for consideration by *eLife*. Your article has been reviewed by three peer reviewers, including Margaret M McCarthy as the Reviewing Editor and Reviewer #1, and the evaluation has been overseen by Catherine Dulac as the Senior Editor.

The reviewers have discussed the reviews with one another and the Reviewing Editor has drafted this decision to help you prepare a revised submission.

Summary:

Tamoxifen is a front line therapy for patients diagnosed with and treated for breast cancer, with many women taking the drug for up to 10 years. Non-compliance is often associated with negative side effects experienced by some women. Relatively little attention is paid to how tamoxifen acts on the brain and this report addresses that directly and specifically by exploiting the power of single cell RNA sequencing of the hypothalamus in both the wildtype and a conditional KO of estrogen receptor alpha mouse. This study was designed to identify neurobiological mechanisms underlying the effects of tamoxifen. Mice were given tamoxifen daily for four weeks. Tamoxifen increased indices of heat dissipation and bone density, while decreasing activity. Tamoxifen also produced complex and widespread changes in gene expression in various cell types, as measured by single cell Drop-seq and gene set enrichment analyses. In neurons, more genes were repressed than induced. These changes were largely ablated by tamoxifen in mice with specific knockout of ERα in the hypothalamus, suggesting that tamoxifen-induced repression of gene expression is mediated by ERα. ERα KO also blocked effects of tamoxifen on heat dissipation, bone density, and activity, suggesting by extension that tamoxifen's affects these variables in least in part via ERα-mediated gene expression.

1) A major theoretical concern was raised by one reviewer: Whether all this detailed observations have any relevance for the human condition is absolutely unknown. In this regard, it would be important to clarify what the authors mean by "tamoxifen therapy"? Of human or of mice? If of humans (which seems to be the scenario), this clearly is not possible to test in mice. Where this work may be relevant actually is the use of tamoxifen in transgenic construct of mouse models. However, the tamoxifen intervention did not mimic that approach. Thus, in the end, it is unclear what this extensive data set reflects regarding the human condition or other issues. It appears the authors are clearly aiming to address Tamoxifen's side effects regarding the human condition. What does it mean to state there are "side-effect" regarding mice? They should change the wording on the specifics as it relates to mice but could discuss the issue in the Discussion as a potential relevance to humans.

2) The gene expression data are impressive and have the potential to identify specific genes modulated by tamoxifen in an ERα-dependent manner. However, a more explicit description of the key genes in neurons that are differentially expressed in wild-type and ERα KOs in response to tamoxifen would be very useful. This information wasn't clear from the supplementary files, and the volcano plots, while providing pretty Rorschach-like pictures of differential gene expression, provide no specifics about the identity of the genes that are differentially regulated by tamoxifen in wild-type and ERα KOs. From the perspective of drug development, which the authors emphasized in the Introduction, what DEGs would be worth targeting to mitigate the side effects of tamoxifen? Without information about specific DEGs, it's hard to see how the Drop-seq data provide significant insights that could lead to new adjuvant therapies to prevent tamoxifen side effects.

3) To better understand the effects of Erα KO on gene expression, it would be useful to mention somewhere in the paper the relative abundance and distribution of hypothalamic ERα in the cell types examined (particularly neurons and the various glia types). Do hypothalamic ERβ, GPER, or even PRs contribute to thermoregulation, bone density, or activity and if so, would the authors expect similar interactions with tamoxifen if these receptors were knocked out?

4) Although the data support that tamoxifen reduces core body temperature and increases tail temperature over the course of many hours during the light cycle, hot flashes in women are characterized by sudden increases in temperature that dissipate within a few minutes. Thus, the authors should be careful in equating these longer-term changes to the more temporally distinct hot flashes and acknowledge this caveat in the Discussion.

5) “Single-cell expression analysis was unable to detect a significant decrease of *Esr1* expression in any cell type of *Esr1^cKO^* mice, but clearly demonstrated that the NK2 homeobox transcription factor *Nkx2-1* was enriched in neurons and in ependymal cells” – if I am understanding this correctly it means there is no evidence that *Esr1* was in fact deleted in neurons and ependymal cells. This seems problematic, can the authors provide more of an explanation and some assurances that the model is indeed what they say it is?

6) “29,807 cells (8,220 from n = 3 vehicle treated mice, 21,587 from n = 5 tamoxifen treated mice)” – can the authors address whether it is problematic that they have almost 3 times as many cells from tamoxifen treated mice as controls.

7) It wasn't clear how Figure 5—figure supplement 3 supports the statement, "Together these data indicate that hypothalamic expression of ERα is required for a large proportion of the gene expression changes normally induced by tamoxifen, including many functional pathways and those controlling neuronal activity and estrogen targets in many cell types and neuronal subtypes.". The graphs seem to indicate little overlap between wild-type and *Esr1* KO.

Title:

In light of the comments of the reviewer, the authors might consider removing "side effects" from the title.

Revisions expected in follow-up work:

Revisions needed are addressed above and are largely along the lines of clarification and reframing the findings in terms of whether they are or are not clinically relevant

---

## [Author Response]

Revisions for this paper:1) A major theoretical concern was raised by one reviewer: Whether all this detailed observations have any relevance for the human condition is absolutely unknown. In this regard, it would be important to clarify what the authors mean by "tamoxifen therapy"? Of human or of mice? If of humans (which seems to be the scenario), this clearly is not possible to test in mice. Where this work may be relevant actually is the use of tamoxifen in transgenic construct of mouse models. However, the tamoxifen intervention did not mimic that approach. Thus, in the end, it is unclear what this extensive data set reflects regarding the human condition or other issues. It appears the authors are clearly aiming to address Tamoxifen's side effects regarding the human condition. What does it mean to state there are "side-effect" regarding mice? They should change the wording on the specifics as it relates to mice but could discuss the issue in the Discussion as a potential relevance to humans.

This point is well taken. The revision now reserves the use of the word “therapy” for when discussing tamoxifen therapy only in humans. To make this distinction clear, and to appropriately limit interpretations of our data, we have changed the title to “Estrogen receptor alpha in the brain mediates tamoxifen induced changes in physiology in mice” and removed the word “therapy” from our experimental design, reporting of results, and the majority of the Discussion. We do discuss tamoxifen therapy in humans when setting up the problem in the Introduction and again at the end of the Discussion to highlight a potential relevance to humans, as suggested.

2) The gene expression data are impressive and have the potential to identify specific genes modulated by tamoxifen in an ERα-dependent manner. However, a more explicit description of the key genes in neurons that are differentially expressed in wild-type and Erα KOs in response to tamoxifen would be very useful. This information wasn't clear from the supplementary files, and the volcano plots, while providing pretty Rorschach-like pictures of differential gene expression, provide no specifics about the identity of the genes that are differentially regulated by tamoxifen in wild-type and Erα KOs. From the perspective of drug development, which the authors emphasized in the Introduction, what DEGs would be worth targeting to mitigate the side effects of tamoxifen? Without information about specific DEGs, it's hard to see how the Drop-seq data provide significant insights that could lead to new adjuvant therapies to prevent tamoxifen side effects.

We thank the reviewer for these important clarifying comments about the Drop-seq data. Our initial submission was focused on the general effects of tamoxifen on gene expression in the different cell types examined. As stated in the text, this indicates that neurons and ependymal cells are most transcriptionally responsive to tamoxifen administration. We went on to highlight pathways (neuropeptide signaling, oxidative phosphorylation, glycolysis) that are important for neuronal function and are coordinately downregulated by tamoxifen in wild-type neurons but not in cKO neurons. Nevertheless, we can see how a lack of individual gene names could seem obfuscating or unsatisfying.

For the pathway analyses, we now highlight genes in Figure 4C, identifying individual genes within the significant pathways that may be potential candidates for follow up studies. Furthermore, we reasoned that if individual differentially expressed genes are potential regulators of tamoxifen’s physiological effects, then they are likely to be discordantly regulated in the wild-type vs. cKO hypothalamus-POA. We now provide a complete list of genes that are significantly (adj. *p*<0.05) differentially expressed in wild-type cells treated with tamoxifen vs. vehicle (Supplementary file 1I and J). The supplementary files also provide corresponding differential expression data for these same genes in cKO cells. Although this information is available in the supplementary files, we highlighted some of these genes in Figure 5D, which simultaneously illustrates how individual genes are regulated by tamoxifen in wild-type cells and cKO cells.

3) To better understand the effects of ERα KO on gene expression, it would be useful to mention somewhere in the paper the relative abundance and distribution of hypothalamic ERα in the cell types examined (particularly neurons and the various glia types). Do hypothalamic ERβ, GPER, or even PRs contribute to thermoregulation, bone density, or activity and if so, would the authors expect similar interactions with tamoxifen if these receptors were knocked out?

We agree it is important to understand if PR or other ERs contribute to the changes induced by tamoxifen. The relative abundance and distribution of hypothalamic ERα in the various cell types is highlighted in Figure 3B, which shows the percent of cells within each cell type in which we detected *Esr1, Esr2, Pgr,* and *Gper1*. The distribution of *Esr1, Esr2, Pgr,* and *Gper1* are also now provided for neuronal subtypes in Figure 4B (these are now included as the first four genes on the x-axis).

Although we have only examined the effect of ERα conditional KO on body temperature, bone density and physical activity, it is possible that ERα expressed outside of the *Nkx2-1* lineage or other receptors beyond than ERα may also be involved. The revised manuscript acknowledges this possibility in several sections of the manuscript, as outlined below:

a) We examine *Esr1, Esr2, Pgr,* and *Gper1* expression in our data (Figure 3B, Figure 4B).

b) We cite studies that show effects of other receptors on thermoregulation in the Discussion: Opas et al., 2009, Roepke et al., 2010), physical activity: Ogawa et al., 2003), and bone: Windahl et al., 2002, Lindberg et al., 2002, Roepke et al., 2010).

c) This more comprehensive evaluation of our data and the literature sets up the conclusion that “ERα signaling within the *Nkx2-1* lineage as a major mediator” (not necessarily the only mediator) “of the effects of tamoxifen on thermoregulation, physical activity, and bone homeostasis”.

4) Although the data support that tamoxifen reduces core body temperature and increases tail temperature over the course of many hours during the light cycle, hot flashes in women are characterized by sudden increases in temperature that dissipate within a few minutes. Thus, the authors should be careful in equating these longer-term changes to the more temporally distinct hot flashes and acknowledge this caveat in the Discussion.

Thank you for pointing out this important distinction. Along with the distinction between tamoxifen therapy in humans and tamoxifen treatment in mice, any effects on hot flashes are quite distinct from effects on thermoregulation. (We think that vasomotor symptoms in women are a consequence of changes in thermoregulation that accompany menopause but that is not the point of the current manuscript.) We have acknowledged this caveat in the Discussion and drawn parallels between our findings and the effects of ovariectomy in rodents, which also leads to higher tail temperature and sometimes lower core temperature.

“However, it is important to acknowledge that the sustained increase in tail skin temperature observed here and in other animal models (Kobayashi et al., 2000, Padilla et al., 2018) does not fully reflect the episodic nature of hot flashes in humans.”

5) “Single-cell expression analysis was unable to detect a significant decrease of Esr1 expression in any cell type of Esr1^cKO^ mice, but clearly demonstrated that the NK2 homeobox transcription factor Nkx2-1 was enriched in neurons and in ependymal cells” – if I am understanding this correctly it means there is no evidence that Esr1 was in fact deleted in neurons and ependymal cells. This seems problematic, can the authors provide more of an explanation and some assurances that the model is indeed what they say it is?

As the reviewer astutely pointed out, we unfortunately did not detect a significant reduction of *Esr1* transcript in any cell types. A noted limitation of scRNA-seq is the loss of statistical power as cells are divided into individual clusters and analyzed. This loss of power particularly affects analysis of low expression transcripts like *Esr1*. Therefore, we examined ERα protein expression by immunohistochemistry in wild-type and cKO mice, which appears in Figure 5A. Indeed, ERα immunoreactivity is absent in the VMH, ARC, and POA of the conditional KO model, as shown in Figure 5A and in previous publications from our group (Correa et al., 2015, Herber et al., 2019). Nevertheless, we agree that showing the gene expression data for *Esr1* is important, despite the fact that we observed no significant decrease. We have now added this analysis as Figure 5—figure supplement 1A as fold changes in *Esr1* expression in cKO compared to wild-type mice, showing essentially no change.

6) “29,807 cells (8,220 from n = 3 vehicle treated mice, 21,587 from n = 5 tamoxifen treated mice)” – can the authors address whether it is problematic that they have almost 3 times as many cells from tamoxifen treated mice as controls.

This is a very important point, and we thank the reviewer for highlighting it. The imbalance in the number of cells was a consequence one control mouse dying just before completion of the treatment timeline. With the extra space in our drop-seq pipeline, we included an extra tamoxifen-treated mouse that was treated concurrently and available at the time. Like the reviewer, we were trepidatious about analyzing an unequal number of vehicle and control mice, however, the authors of the scRNA-seq package that we used (Seurat) claim that imbalanced cell numbers are not inherently a problem (https://github.com/satijalab/seurat/issues/1325).

Nevertheless, to address this concern that we share with the reviewer, we randomly sampled our groups after initial quality filtering such that each group would contain 8,220 cells. As shown in Author response images 1 and 2, the results of the down-sampled analyses mirror the results found in the initial submission with respect to: Cell types identified, distribution of ERs, the unique sensitivity of neurons and ependymal cells to tamoxifen, the effect of tamoxifen on functional pathways, and the discordant effects of tamoxifen on wild-type and cKO cells. Together these data indicate that the imbalance in cell number did not affect the outcome of the analyses, and so we believe that it is appropriate to leave the initially submitted analysis, which has more statistical power, in the manuscript.

**Author response image 1. sa2fig1:** Downsampled re-analysis of scRNA-Seq: all treatment groups normalized to exactly 8,220 cells after initial quality filtering.

7) It wasn't clear how Figure 5—figure supplement 3 supports the statement, "Together these data indicate that hypothalamic expression of ERα is required for a large proportion of the gene expression changes normally induced by tamoxifen, including many functional pathways and those controlling neuronal activity and estrogen targets in many cell types and neuronal subtypes.". The graphs seem to indicate little overlap between wild-type and Esr1KO.

We thank the reviewer for pointing out the confusing manner in which we described our data. As the reviewer correctly points out, there is little overlap between wild-type and *Esr1*cKO cells in response to tamoxifen, and this is the point we intended to convey. The Results section explaining Figure 5 and its supplements has been re-written for clarity. Furthermore, labeling in Figure 5 and its supplements has been re-worked to convey the point the reviewer makes.

Title:In light of the comments of the reviewer, the authors might consider removing "side effects" from the title.

We have changed the title to “Estrogen receptor alpha in the brain mediates tamoxifen induced changes in physiology in mice”.